**Subject Category:**
Biology (whole organism)

evolution/taxonomy and systematics/ biogeography

ancient lakes, Pontocaspian, evolution lake-level fluctuations, salinity, molluscs

**Author for correspondence:**
Arthur F. Sands
e-mail: arthur.f.sands@allzool.bio.uni-giessen.de

# Old lake versus young taxa: a comparative phylogeographic perspective on the evolution of Caspian Sea gastropods (Neritidae: *Theodoxus*)

Arthur F. Sands[1], Thomas A. Neubauer[1,2], Saeid Nasibi[3], Majid Fasihi Harandi[3], Vitaliy V. Anistratenko[4], Thomas Wilke[1] and Christian Albrecht[1]

[1]Department of Animal Ecology and Systematics, Justus Liebig University Giessen, Heinrich-Buff-Ring 26–32 (IFZ), 35392 Giessen, Germany
[2]Naturalis Biodiversity Centre, PO Box 9517, 2300 Leiden, The Netherlands
[3]Research Center for Hydatid Disease in Iran, School of Medicine, Kerman University of Medical Sciences, Kerman 76169-14115, Iran
[4]I.I. Schmalhausen Institute of Zoology of the National Academy of Sciences of Ukraine, B. Khmelnitsky Street, 15, Kiev 01030, Ukraine

AFS, 0000-0003-0966-421X; TAN, 0000-0002-1398-9941;
SN, 0000-0002-7628-5415; MFH, 0000-0003-3257-5389;
VVA, 0000-0003-0832-7625; TW, 0000-0001-8263-7758;
CA, 0000-0002-1490-1825

The Caspian Sea has been a highly dynamic environment throughout the Quaternary and witnessed major oscillations in lake level, which were associated with changes in salinity and habitat availability. Such environmental pressures are considered to drive strong phylogeographic structures in species by forcing populations into suitable refugia. However, little is actually known on the effect of lake-level fluctuations in the Caspian Sea on its aquatic biota. We compared the phylogeographic patterns of the aquatic Neritidae snail genus *Theodoxus* across the Pontocaspian region with refugial populations in southern Iran. Three gene fragments were used to determine relationships and divergence times between the sampled populations in both groups. A dated phylogeny and statistical haplotype networks were generated in conjunction with the analyses of molecular variance and calculations of isolation by distance using distance-based redundancy analyses. Extended Bayesian skyline plots were constructed to assess demographic history. Compared with the southern Iranian populations, we found little phylogeographic structure for the Pontocaspian *Theodoxus* group, with more recent diversification,

homogeneity of haplotypes across the Pontocaspian region and a relatively stable demographic history since the Middle Pleistocene. Our results argue against a strong influence of Caspian Sea low stands on the population structure post the early Pleistocene, whereas high stands may have increased the dispersal possibilities and homogenization of haplotypes across the Pontocaspian region during this time. However, during the early Pleistocene, a more dramatic low stand in the Caspian Sea, around a million years ago, may have caused the reduction in *Theodoxus* diversity to a single lineage in the region. In addition, our results provide new insights into *Theodoxus* taxonomy and outlooks for regional conservation.

# 1. Introduction

The Caspian Sea is one of the largest and oldest lakes on the planet. Barring episodic overflow events, the Caspian Sea has been an isolated endorheic basin since at least the early Pliocene, around 5.3 million years ago (Ma) [1]. Today, the Caspian Sea and its catchment cover an area of approximately 3 500 000 km$^2$ [2,3]. It is renowned for its dramatic historical lake-level fluctuations and salinity shifts, primarily as a consequence of glacial cycles during the Quaternary [1,4]. Moreover, temporary connections occurred between the Caspian Sea and other Pontocaspian basins (i.e. the Aral, Azov and Black seas [1]) during high stands. These changes are hypothesized to have had a major effect on the evolutionary history of its aquatic biota, such as the highly endemic gastropod fauna [5]. Such fluctuations make the Caspian Sea a suitable model system to study the effect of palaeo-environmental changes on the phylogeographic structure in long-lived lake biota.

The recurrent fluctuations in salinity and lake level during the Quaternary may have created periodic temporal refugia. Isolated Caspian sub-basins during low stands or spring sources and riverine systems during the periods of increased salinity are potential candidate refugia. The restricted gene flow, across such refugial populations, is expected to generate a strong phylogeographic structure within the species as a consequence of isolated evolutionary histories. Compared with the large extent of the Caspian drainage basin, surprisingly only a small number of phylogeographic and population-level studies exist for aquatic taxa. These studies primarily focus on fish and crustaceans in a broader Pontocaspian context (see [6–14]). Interestingly, while many of these studies indicate a strong geographic structure across the entire Pontocaspian region, most show a lack of visible intraspecific structure in the Caspian drainage basin [7,9–11,14]. Although this may be a genuine evolutionary pattern (e.g. due to a lack of genetic breaks), there may be a bias. Often, taxa were selected that are easily dispersed, or the sampling design was rather narrow across the Caspian basin itself.

Gastropods can make good candidates for documenting responses to environmental changes as they often have limited dispersal capabilities, fast generation turnover, good fossil preservation and are sensitive to ecological shifts [15–19]. Selecting a widespread Pontocaspian gastropod taxon, such as the neritid genus *Theodoxus*, which is both dioecious and lacks a planktonic larval stage that facilitates dispersal [20], may not only improve our understanding of the effects of major salinity and lake-level changes on the evolutionary history of Pontocaspian gastropods, but also enhance our understanding of the phylogeography and the existence of refugia in the Caspian system as a whole. Most *Theodoxus* spp. are generalists and occur in lakes, rivers, estuaries and springs, although they depend on well-oxygenated water and environments with a hard substrate, where they can graze on algae (e.g. rocks and stones, shell beds or certain aquatic plants) [20]. Due to the abundance of these environments in the Pontocaspian region, *Theodoxus* has become a common component of its malacofauna [5,21]. Four species have been considered endemic to the Pontocaspian region and share a presence in the Caspian basin; *T. astrachanicus* Starobogatov in Starobogatov *et al.*, 1994, *T. major* Issel, 1865, *T. pallasi* Lindholm, 1924 and *T. schultzii* (Grimm, 1877) (figure 1) [5,22]. Despite considerable phenotypic variability (figure 1), recent studies based on morphological [23] and molecular analyses [22] have led to speculations that these four may represent a single species [5]. Here, we use this group as a model taxon to determine the effects of major limnological events in the Caspian Sea on the phylogeographic structure of Pontocaspian gastropods. We compare their phylogeographic structure with a potential sister group of *Theodoxus* occurring in isolated spring systems in southern Iran, feeding the remnants of endorheic basins (as a model for gastropods that have persisted in refugia). To reach this objective, we (i) sampled *Theodoxus* across the Pontocaspian and southern Iranian drainage networks, (ii) used a molecular dataset to carry out phylogenetic and phylogeographic analyses to determine population structure, and (iii) tested for demographic expansion/contraction events through time, which may help to identify genetic bottlenecks.

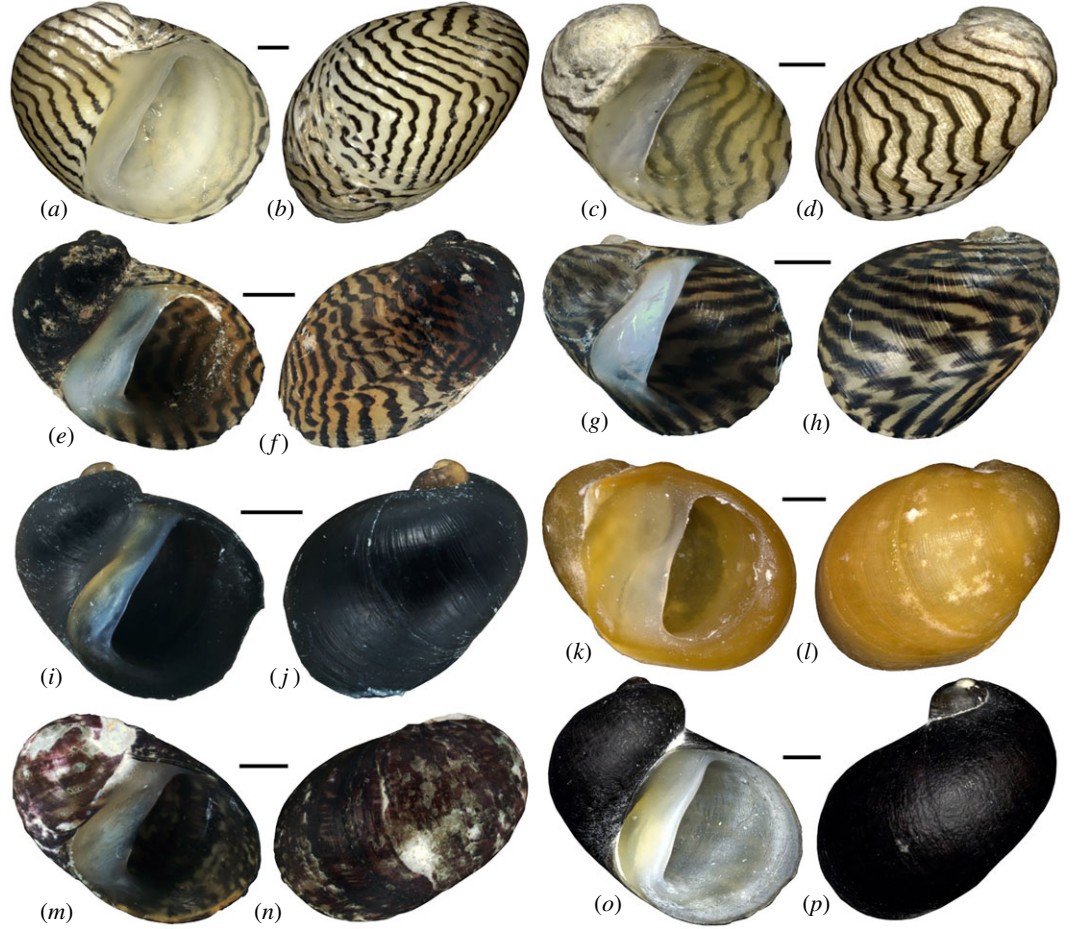

**Figure 1.** Representative phenotypes of the Pontocaspian and southern Iranian *Theodoxus* species studied herein. Pontocaspian: (*a,b*) *T. pallasi* (UGSB 20712); (*c,d*) *T. astrachanicus* (UGSB 18130); (*e,f*) *T. pallasi* (UGSB 18091); (*g,h*) *T. major* (UGSB 20482); (*i,j*) *T. major* (UGSB 20496); (*k,l*) *T. schultzii* (UGSB 20791). Southern Iranian: (*m,n*) *T. doriae* (UGSB 21706); (*o,p*) *T. pallidus* (UGSB 22228). Scale bar, 1 mm.

We hypothesize that repeated Caspian Sea low stands with increased salinity resulted in multiple isolated refugia across the Caspian Sea and its catchments and thus a high degree of population structure. As such, Pontocaspian and Iranian *Theodoxus* should possess similar population structures. Additionally, as the taxonomy of some of these species is poorly resolved, we provide a molecular perspective on the identities of species. The outcomes of this research should identify the timing and extent of how major salinity and lake-level changes may have affected the evolutionary histories of native Pontocaspian aquatic gastropods and identify Pontocaspian refugia (particularly across the Caspian Sea drainage network). Moreover, we discuss how our results may help resolve taxonomic uncertainties and what the implications may mean for the conservation of Pontocaspian taxa.

## 2. Material and methods

### 2.1. Sample collection and laboratory protocols

*Theodoxus* specimens were collected and stored following Sands *et al*. [22]. We included the genetic data of either 5 or 10 specimens per location to allow for robust analyses (table 1 and figure 2). Sands *et al*.'s [22] protocols were again followed to extract and amplify two mtDNA fragments; cytochrome *c* oxidase subunit I (COI) and 16S rRNA (16S) and one nDNA intron fragment, ATP synthetase subunit α (ATPα). Total genomic DNA was extracted from the foot tissue of the snails using a DNeasy Blood and Tissue kit (QIAGEN, Hilden, Germany) and amplified using the primers TheoSF1 and TheoSR1 for COI [22], 16Sar-L and 16Sbr-H for 16S [24] and ATPαSaf1 and ATPαSar1 for ATPα [25]—for primer sequences and PCR conditions, see Sands *et al*. [22]. Purification and bidirectional Sanger sequencing of the amplified gene fragments were carried out by LGC Ltd. (Berlin, Germany). Where needed, we incorporated

**Table 1.** Species details and GenBank accession numbers for Pontocaspian, southern Iranian and outgroup *Theodoxus* spp. Locality names correspond to those in figure 2.

| *Theodoxus* species | number of specimens | locality | country | GPS coordinates | GenBank accession numbers | | | |
|---|---|---|---|---|---|---|---|---|
| | | | | | COI | 16S | ATPα | |
| **Pontocaspian group** | | | | | | | | |
| *T. major* | 10 | Yerevan | Armenia | 40.16633° N, 44.48533° E | MN168547–168556 | MN174926–174935 | MN180417–180426 | |
| *T. major* | 5 | Aknalich | Armenia | 40.14288° N, 44.17117° E | MK754532–754534, MN168557–168558 | MK754874–754876, MN174936–174937 | MK755206–755208, MN180427–180428 | |
| *T. pallasi* | 10 | Blue Planet Beach | Azerbaijan | 40.77513° N, 49.54489° E | MN168559–168568 | MN174938–174947 | MN180429–180438 | |
| *T. pallasi* | 5 | Pirallahi Island | Azerbaijan | 40.489336° N, 50.330422° E | MK754691–754693, MN168569–168570 | MK755030–755032, MN174948–174949 | MK755347–755349, MN180439–180440 | |
| *T. pallasi* | 10 | Masalli | Azerbaijan | 39.01879° N, 48.69972° E | MN168571–168580 | MN174950–174959 | MN180441–180450 | |
| *T. cf. major* | 10 | Baba Aman Spring | Iran | 37.48488° N, 57.43629° E | MN168581–168590 | MN174960–174969 | MN180451–180460 | |
| *T. pallasi* | 5 | Shahdol River | Iran | 36.584867° N, 51.768145° E | MK754724–754725, MK754766, MN168591–168592 | MK755063–755064, MK755105, MN174970–174971 | MK755377–755378, MK755415, MN180461–180462 | |
| *T. cf. major* | 10 | Zoeram Spring | Iran | 37.31885° N, 57.73742° E | MN168593–168602 | MN174972–174981 | MN180463–180472 | |
| *T. pallasi* | 10 | Kuryk | Kazakhstan | 43.183287° N, 51.652672° E | MN168638–168647 | MN175017–175026 | MN180508–180517 | |
| *T. pallasi* | 10 | Aktau | Kazakhstan | 43.628058° N, 51.168252° E | MN168648–168657 | MN175027–175036 | MN180518–180527 | |

**Table 1.** (*Continued.*)

| *Theodoxus* species | number of specimens | locality | country | GPS coordinates | GenBank accession numbers | | | |
|---|---|---|---|---|---|---|---|---|
| | | | | | COI | 16S | ATPα | |
| *T. pallasi* | 10 | Saura Canyon | Kazakhstan | 44.221987° N, 50.806791° E | MN168658–168667 | MN175037–175046 | MN180528–180537 | |
| *T. schultzii* | 5 | Caspian Sea | Kazakhstan | 43.50589° N, 51.08473° E | MN168668–168672 | MN175047–175051 | MN180538–180542 | |
| *T. astrachanicus* | 10 | Damchik | Russia | 45.78942° N, 47.88781° E | MN168673–168682 | MN175052–175061 | MN180543–180552 | |
| *T. astrachanicus* | 10 | Astrakhan | Russia | 46.34941° N, 48.01978° E | MN168683–168692 | MN175062–175071 | MN180553–180562 | |
| *T. astrachanicus* | 10 | Selitrennoye | Russia | 47.16708° N, 47.44868° E | MN168693–168702 | MN175072–175081 | MN180563–180572 | |
| *T. astrachanicus* | 10 | Volgograd | Russia | 48.42905° N, 44.94628° E | MN168703–168712 | MN175082–175091 | MN180573–180582 | |
| *T. astrachanicus* | 10 | Volga-Don Canal | Russia | 48.4541° N, 44.36181° E | MN168713–168722 | MN175092–175101 | MN180583–180592 | |
| *T. astrachanicus* | 10 | Rostov-on-Don | Russia | 47.18499° N, 39.62985° E | MN168723–168732 | MN175102–175111 | MN180593–180602 | |
| *T. astrachanicus* | 10 | Utlyukskij Liman A | Ukraine | 46.2857° N, 35.29080° E | MN168733–168742 | MN175112–175121 | MN180603–180612 | |
| *T. astrachanicus* | 5 | Utlyukskij Liman B | Ukraine | 46.14998° N, 35.04865° E | MN168743–168747 | MN175122–175126 | MN180613–180617 | |

(*Continued.*)

**Table 1.** (Continued.)

| Theodoxus species | number of specimens | locality | country | GPS coordinates | GenBank accession numbers | | | |
|---|---|---|---|---|---|---|---|---|
| | | | | | COI | 16S | ATPα | |
| **Southern Iranian group** | | | | | | | | |
| T. doriae | 10 | Shahrbabak | Iran | 30.11875° N, 55.12171° E | MN168603–168612 | MN174982–174991 | MN180473–180482 | |
| T. doriae | 10 | Harat | Iran | 30.01509° N, 54.34030° E | MN168613–168622 | MN174992–175001 | MN180483–180492 | |
| T. pallidus | 10 | Qatruyeh | Iran | 29.169889° N, 54.684617° E | MN168623–168632 | MN175002–175011 | MN180493–180502 | |
| T. doriae | 5 | Poshtekeno Spring | Iran | 27.82° N, 56.40° E | MN168633–168637 | MN175012–175016 | MN180503–180507 | |
| T. pallidus | 10 | Aspas | Iran | 30.66724° N, 52.29326° E | MN168748–168757 | MN175127–175136 | MN180618–180627 | |
| **Outgroups** | | | | | | | | |
| T. jordani | 1 | Bath of Aphrodite | Cyprus | 35.056477° N, 32.346027° E | MK754676 | MK755015 | MK755332 | |
| T. transversalis | 1 | Vardar River | Macedonia | 41.1504° N, 22.52371° E | MK754769 | MK755108 | MK755416 | |

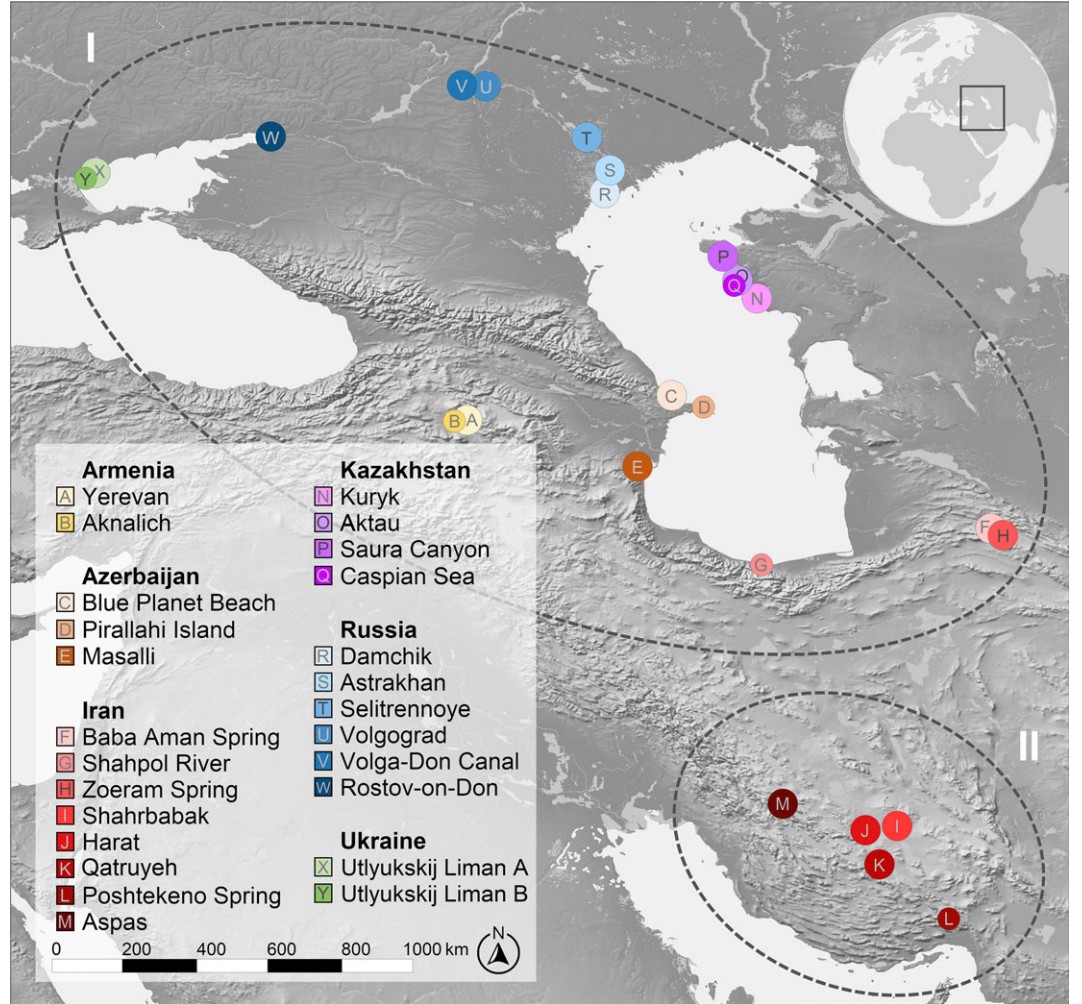

**Figure 2.** Map depicting the locations of the sampling sites around the Pontocaspian system and southern Iran. Colours of dots correspond to the locations, as indicated in the key. Dashed lines encircle (I) the Pontocaspian and (II) the southern Iranian *Theodoxus* sampling localities. The size of the dots represents the sample size at each location (larger = 10 specimens; smaller = 5 specimens).

published GenBank sequences of individuals where all three DNA fragments were available. Sequence ends were trimmed and aligned in Genious 10.1.2 [26] using the Genious alignment algorithm. To test for saturation over 1000 replicates [27], we used DAMBE 5.0.52 [28,29]. None of the three molecular datasets showed any significant degree of saturation (COI, 16S and ATPα; $p < 0.001$).

## 2.2. Phylogenetic analyses

To investigate the relationships among all *Theodoxus* specimens, delimit phylogroups and provide divergence date estimates between and within the Pontocaspian and southern Iranian groups, a dated Bayesian phylogeny was constructed. The phylogeny combined all amplified gene fragments (1609 bp in total) for 222 individuals (211 sequenced for the first time; table 1).

First, a lognormal relaxed clock was set for all gene datasets and a birth–death tree prior was selected. bModelTest v. 1.1.2 [30], as implemented in BEAST v. 2.5.2 [31], was used to determine the best-fit model for each gene set. TN93 (121 131) was determined the optimal model for COI and ATPα and a variant of the HKY (with an additional group for the rates $r_{ct}$ and $r_{gt}$; 121 323) for 16S. Second, sequences derived from *T. jordani* (Sowerby, 1836) and the more distantly related *T. transversalis* (Pfeiffer, 1828) were used to root the phylogeny (see [22]). As fossil dating is challenging for *Theodoxus*, the phylogeny was calibrated using the molecular clock rates and secondary dating of internal nodes, as established by Sands *et al.* [22]. For congruency, clock rates and secondary dates were set to the 95% confidence intervals and highest posterior densities (HPD) established therein. Molecular clock rate priors were linearly distributed (COI, 1.78%, standard deviation (s.d.) 2.18–1.40%; 16S, 0.73%, s.d. 0.98–0.48%; ATPα, 0.65%, s.d. 0.95–0.37%), while the secondary dating priors were set to normal distribution (*T. transversalis* from all

specimens, 8.27 Ma, 95% HPD 11.60–4.90 Ma; *T. jordani* from all in-group specimens, 4.77 Ma, 95% HPD 6.33–3.20 Ma).

Thereafter, two independent runs of 200 000 000 MCMC generations, saving one tree in every 20 000 generations, were constructed in BEAUti v. 2.5.2 [31] and implemented in BEAST v. 2.5.2 through the CIPRES Science Gateway [32]. LogCombiner v. 2.5.2 [31] was used to combine trees and log files of each run with 50% burn-in removed. Validation of the convergence and mixing of the combined log file was assessed in Tracer v. 1.7.1 [33] to ensure that all effective sample size (ESS) values were greater than 200 and TreeAnnotator v. 2.5.2 [31] was used to summarize trees, with no further burn-in removed. Phylogroups were delimited based on posterior probability (PP) support values, where the most recent daughter lineages stemming from a common ancestor were both significantly supported (PP $\geq$ 0.95).

## 2.3. Phylogeographic structure

Evolutionary relationships, among COI, 16S and ATPα haplotypes of the Pontocaspian and southern Iranian *Theodoxus* groups, were established independently for each group and gene through statistical haplotype networks in TCS v. 1.21 [34]. Sequence ambiguities associated with heterozygote states in the ATPα sequences (nDNA) were resolved by determining 90% probability score alleles in Phase v. 2.1 [35] as implemented in DnaSP v. 6.11.01 [36] under default settings. To further test for differentiation within each group, among geographical sampling localities, analyses of molecular variance (AMOVAs) were performed in Arlequin v. 3.5.2.2 [37]. *p*-values were subjected to Holm's sequential Bonferroni corrections [38]. Furthermore, Arlequin was used to calculate haplotypic (*h*) and nucleotide diversities (*π*). Finally, isolation by distance (IBD) was tested for using the distance-based redundancy analysis (db-RDA) [39,40] with the package vegan v. 2.5-4 [41] in the R statistical environment v. 3.5.2 [42]. Since longitude, latitude and geographical distance among localities were strongly correlated, we only included the latter parameter in the analysis. Principal coordinates analysis was performed on the matrices of pairwise geographical distances and the first principal component was used as single continuous variable in the db-RDA [43]. All input matrices of genetic and geographical distance were constructed in GenAlEx v. 6.5 [44].

## 2.4. Evaluation of demographic history

To provide a temporal perspective on population demographics, extended Bayesian skyline plots (EBSPs) were constructed independently for the pooled Pontocaspian and southern Iranian *Theodoxus* groups. To eliminate the potential bias of our interpretations, caused by individual phylogroups or sampling strategy on the overall trends [45,46], EBSPs were also constructed for individual phylogroups (electronic supplementary material, Appendix A, figure S1). To generate the needed input log files, the BEAST v. 2.5.2 package and CIPRES Science Gateway were again used. The settings and process followed similarly to the dated phylogeny described above (see Phylogenetic analyses). However, 'Coalescent Extended Bayesian Skyline' was set at the tree prior, only in-group taxa of each group were used, and each EBSP was generated through a single run of 800 000 000 MCMC generations (saving one tree and log in every 20 000 generations and one EBSP log in every 5000 generations). The best-fit models were again determined by bModelTest (Pontocaspian group: COI = TN93, 16S = a variant of K81 with additional groups for the rate $r_{gt}$ (123 323), ATPα = a variant of TIM with additional groups for the rates $r_{gt}$ (123 343); southern Iranian group: COI = a variant of HKY with additional groups for the rate $r_{gt}$ (121 123), 16S and ATPα = TN93). Dating followed a slightly different process: crown nodes of each in-group were set according to 95% HPD intervals determined for the date of divergence of the most common recent ancestor (MCRA), as established in the newly generated phylogeny below using a normally distributed prior (figure 3). Log files were assessed in Tracer v. 1.7.1 and only runs where all ESS values were greater than 200 were converted to EBSPs. To construct the EBSPs, the output EBSP log files were imported into the BEAST package tool EBSPAnalyzer v. 2.5.2 [31]. Linear reconstruction was selected and the burn-in removed was set as determined by the parameter convergence in Tracer v. 1.7.1.

# 3. Results

## 3.1. Phylogenetic structure

Based on the combined analyses, the monophyly of Pontocaspian and southern Iranian *Theodoxus* groups was well supported (PP = 1.00; figure 3). Within the two groups, six phylogroups were identified (four in

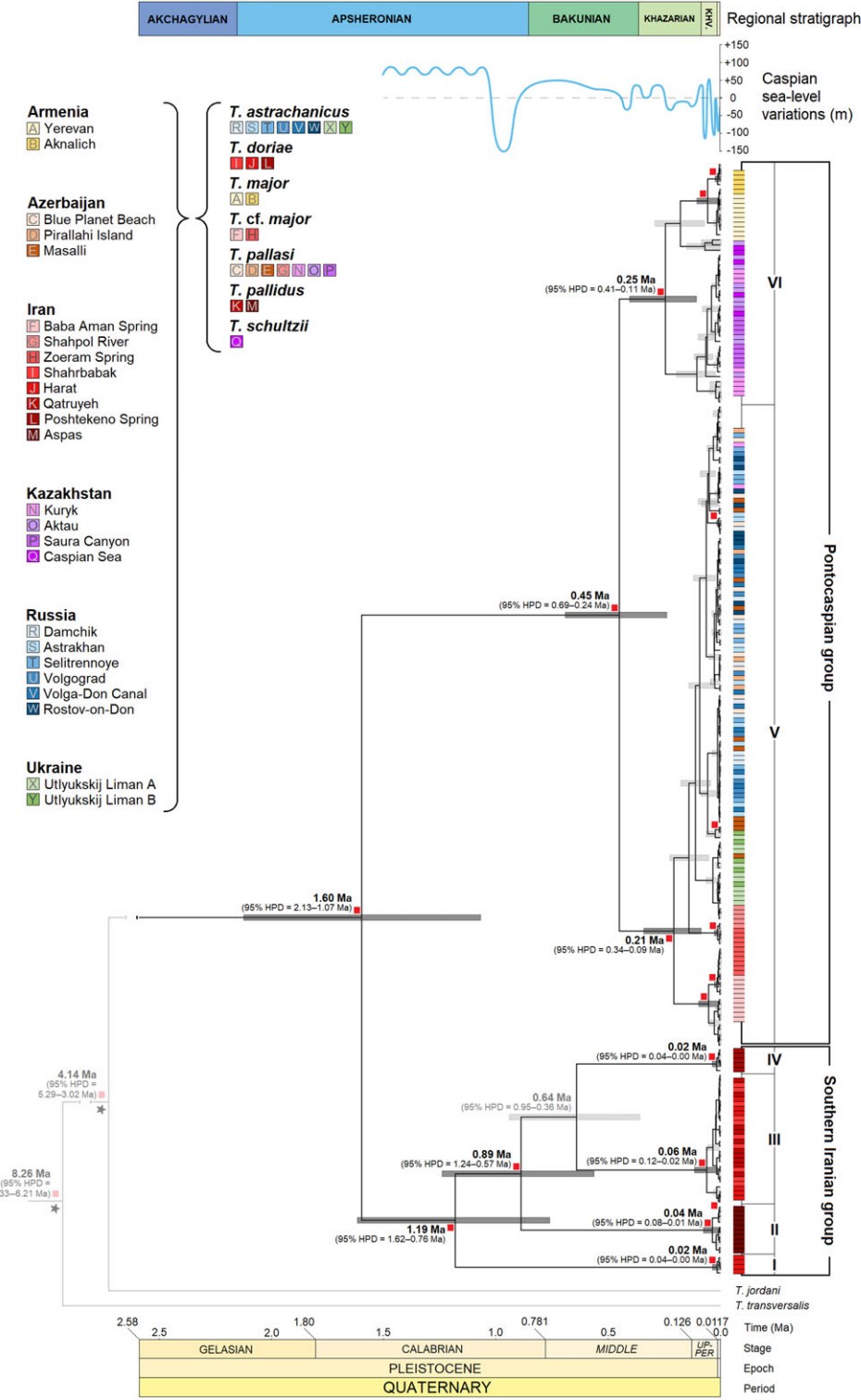

**Figure 3.** Dated phylogeny of Pontocaspian and southern Iranian *Theodoxus* spp. constructed in BEAST based on COI, 16S and ATPα sequence data. Supported phylogroups of Pontocaspian and southern Iranian *Theodoxus* are labelled I to VI. Node labels among these phylogroups and outgroup species denote divergence time in millions of years ago (Ma), with the 95% credibility interval given in parentheses and as grey bars for in-group taxa. Small red squares at nodes (with darkened node bars and, in some instances, dates) indicate significant posterior probabilities of divergence events. Parallel to each supported phylogroup, coloured bars indicate the localities and respective morphospecies of the included specimens as defined in the key on the left. Caspian Sea lake-level variations over the last 1.5 million years (relative to absolute sea level) and regional stratigraphy (following the 'short–Akchagylian' option) are adapted from Krijgsman *et al.* [1] (Khv., Khvalynian).

**Table 2.** Comparative summary of population statistics for COI, 16S and ATPα datasets between Pontocaspian and southern Iranian *Theodoxus* groups. Note IBD was calculated using db-RDA (see §2.3).

| population statistics | | Pontocaspian group | | | Southern Iranian group | | |
|---|---|---|---|---|---|---|---|
| | | COI | 16S | ATPα | COI | 16S | ATPα |
| number of specimens | n | 175 | 175 | 175 | 45 | 45 | 45 |
| number of sequences | *n* | 175 | 175 | 350 | 45 | 45 | 90 |
| haplotypic diversity | *h* | 0.753 | 0.706 | 0.791 | 0.680 | 0.610 | 0.812 |
| nucleotide diversity | $\pi$ | 0.007 | 0.002 | 0.003 | 0.018 | 0.007 | 0.006 |
| isolation by distance (IBD) | $R^2$ | 0.001 | 0.008 | 0.001 | 0.180 | 0.474 | 0.058 |
| | *p*-value | 0.089 | <0.000 | <0.001 | <0.001 | <0.001 | <0.001 |

southern Iran, termed I–IV herein, and two in the Pontocaspian system, V–VI; figure 3). The phylogeny suggests that the divergence of southern Iranian phylogroups from one another began earlier than that of the two Pontocaspian phylogroups (figure 3). Except for phylogroup III (with specimens from Harat, Shahrbabak, Qatruyeh), southern Iranian phylogroups are restricted to single localities. However, there are both polyphyly and some paraphyly between the morphospecies *T. pallidus* (Dunker, 1861) and *T. doriae* Issel, 1865 (table 2 and figure 3). In comparison, while there are haplotypes confined to specific locations among the Pontocaspian *Theodoxus*, there are no location-specific phylogroups. Both supported phylogroups contain individuals from various locations and similarly suggest paraphyly and polyphyly among morphospecies commonly identified as *T. major*, *T. schultzii*, *T. pallasi* and *T. astrachanicus*, both within and between phylogroups (table 1 and figure 3).

## 3.2. Phylogeographic structure

TCS haplotype networks indicate less differentiation between haplotypes and more haplotype sharing among the Pontocaspian localities when compared with those from southern Iran (figure 4). Haplotypic diversity (*h*) is high and similar for both groups, but on average, it is marginally higher for the Pontocaspian group (table 2). In comparison, nucleotide diversity (*n*) is far lower in the Pontocaspian group when compared with southern Iran, indicative of a younger flock (table 2). Additionally, both the networks and diversity assessments demonstrate relatively similar levels of *h* for the Pontocaspian group across the different genetic datasets, while the ATPα dataset has a substantially greater *h* than the two mtDNA datasets for the Iranian group (table 2 and figure 4). This can be indicative of population bottlenecking in southern Iran. The AMOVA analyses of each gene set indicate significant levels of regional population differentiation among some more distantly connected Pontocaspian localities (electronic supplementary material, Appendix A, table S1). Similar AMOVA results were also found for southern Iran, although the results of the independent DNA datasets were more conserved (electronic supplementary material, Appendix A, table S2). The db-RDA found significant IBD within both groups; however, in certain tests it could not be either supported (i.e. the COI dataset) or was overall far weaker among Pontocaspian localities (table 2; electronic supplementary material, Appendix B).

## 3.3. Demographic history

The EBSPs suggest the effective population size (in respect of the generation time) in the Pontocaspian group has remained stable until very recently (*ca* 5 thousand years ago (ka)), while the southern Iranian group entered a steady demographic decline earlier (around 90 ka) (figure 5). Individual EBSPs for the different phylogroups (I–VI; figure 3) did not all find suitable ESS values to warrant construction. The EBSPs for the Pontocaspian phylogroups were supported (electronic supplementary material, figure S1) and they show strong similarity to the pooled EBSP for the Pontocaspian group in figure 5. This suggests that individual phylogroups have had limited prejudice against the overall trends established for the Pontocaspian group (figure 5) and combining data has only aided to increase the support. Conversely, EBSPs for the individual Iranian phylogroups lacked ESS support to warrant construction (potentially as a result of limited samples sizes in the respective phylogroups). Although the pooled EBSP for the Iranian group is in agreement with the network and diversity

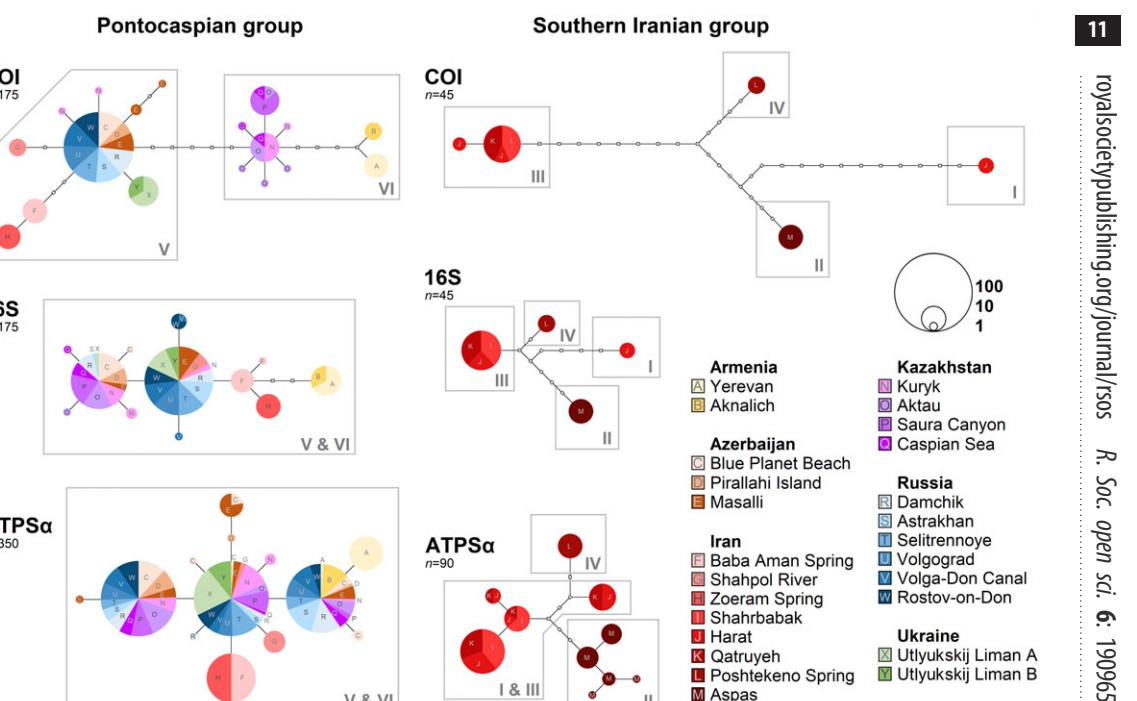

**Figure 4.** Statistical haplotype networks for COI, 16S and ATPα sequence data for Pontocaspian and southern Iranian *Theodoxus* groups. The total number of sequences in each network is demarcated by '*n*'. The circle sizes represent the relative frequency of sequences per haplotype. The number of site changes separating haplotypes is indicated by blank dots. Colours correspond to the sampling locations, as indicated in the key and in figure 2. Haplotype groupings are boxed and labelled according to the phylogroups determined through the dated phylogeny (I–VI; figure 3).

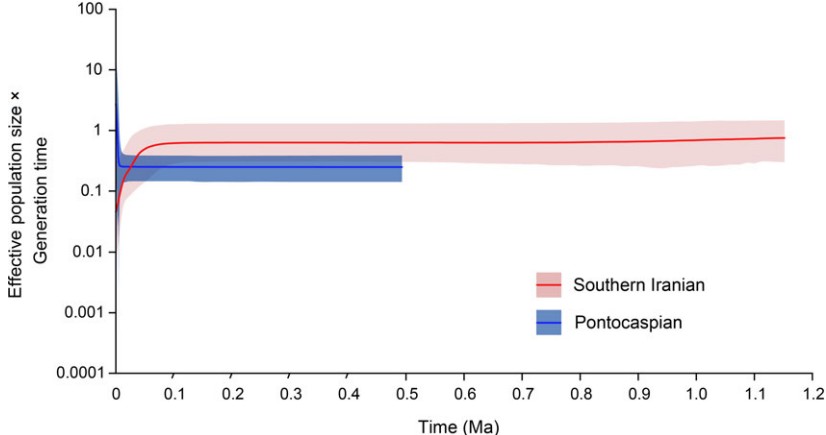

**Figure 5.** EBSPs indicating population trends for the pooled Pontocaspian (blue) and southern Iranian (red) *Theodoxus* groups. The central line of each plot represents the median value and the shaded area indicates the 95% confidence interval. Note the EBSPs depict marginally different starting dates for each group when compared with the phylogeny. Importantly, however, there is a strong overlap of EBSP starting dates with the 95% HPDs established for the onset of intraspecific diversification in each group, as shown in the phylogeny (figure 3).

assessments concerning a possible bottleneck, the plot should be treated with some caution due to lack of being able to confirm this trend across individual phylogroups (figure 5).

# 4. Discussion

## 4.1. Pontocaspian biogeography

We found some indication for an early catastrophic bottleneck reducing *Theodoxus* diversity to a single lineage in the Pontocaspian region during the early Pleistocene. However, post this proposed

bottleneck, there is little intraspecific structure within the Pontocaspian *Theodoxus* group to warrant divergence events being specifically linked to environmental changes (figures 3–5). In contrast with strong intraspecific phylogeographic structure and IBD observed in the southern Iranian *Theodoxus*, haplotypes belonging to the Pontocaspian group are shared among a variety of locations and IBD was either far weaker or not supported at all (table 2 and figures 3 and 4). Therefore, our hypothesis that major low stands drove a strong phylogeographic structure in the Pontocaspian *Theodoxus* group (as a result of populations that were confined to isolated refugia) is inconclusive prior to the Middle Pleistocene, but can be rejected from this point onward.

Supported intraspecific divergence events in Pontocaspian *Theodoxus* do not necessarily correspond with Caspian Sea low stands. Although the 95% confidence intervals on divergence events overlap with several low stands [1], they remain too broad to be attributed to a specific lake level or event (figure 3). Moreover, the EBSP indicates a relatively stable population post the mid-Bakunin until at least the Novo Caspian stage (Holocene; figure 5). If increased salinity or low stands had indeed driven species into refugia, we would have expected to see indications of population bottlenecks during this period. Perhaps Pontocaspian species (or *Theodoxus* in particular) are more tolerant to such environmental changes than expected or the changes during this period were not dramatic enough (both in duration and in intensity) to cause major bottlenecking and force taxa into refugia. A number of Caspian relict molluscs persist in the Caspian Sea [5,47–50], which would suggest it has not been totally inhospitable. Furthermore, some Pontocaspian molluscs have recently been documented to have wide environmental tolerances [5,51,52]. Some *Theodoxus* spp. (including *T. pallasi*) can withstand salinities around 20 psu [20,53,54].

The weak phylogeographic structure observed in the Pontocaspian *Theodoxus* group is also found among other aquatic taxa in the Caspian Sea [6,8,12,14,55], although some are more easily dispersed. Our findings do not mitigate the role lake level and salinity may have in general, but suggest low stands may have had little effect post the mid-Bakunin on Pontocaspian endemics. Moreover, salinity values during low stands probably did not exceed the tolerance of *Theodoxus*. However, Caspian Sea high stands probably increased dispersal possibilities and promoted the homogenization of haplotypes across the Pontocaspian region. As generalists, geographical barriers to *Theodoxus* dispersal within the Caspian Sea are limited, other than by the extreme deep water anoxic environments [5,56,57]. High stands may have increased not only deep water anoxic environments, but also the littoral and sublittoral zones [1], broadening dispersal pathways. This would explain the overlapping and well-distributed phylogroups in the system (phylogroups V and VI; figures 3 and 4). Certain Caspian Sea high stands have also been linked to overflow events into the Sea of Azov [1] and would explain the presence of the phylogroup V in the Utlyukskij Liman (figures 2 and 3).

Although difficult to test, a Caspian Sea low stand may still have had an effect earlier towards the end of the Apsheronian (early Pleistocene). Fossils closely mimicking extant Pontocaspian *Theodoxus* species have been recorded from Apsheronian deposits in Azerbaijan and Turkmenistan [47]. Given this fossil record, one would have expected multiple older lineages in the Pontocaspian (similar to those seen in the southern Iranian group; figure 3) and thus greater levels of nucleotide and haplotypic diversity. However, only a single lineage is present at the start of the Bakunin and diversity indices are indicative of a young flock (table 2 and figure 3). A potential bias from a poorly calibrated molecular clock is unlikely, since the rates derived from Sands *et al.* [22] were based on a well-constrained fossil record. Perhaps a population bottleneck event occurred before the onset of the here-documented diversification, leading to a reboot of Caspian diversity from a single lineage/refugium in the Bakunin. A potential candidate event is the Tyurkyanian regression (1.05–0.95 Ma) at the end of the Apsheronian [1,58]. This low stand, around 150 m below sea level and a temporal extent of *ca* 100 ka, was one of the most severe regressions in the history of the Caspian Sea [1,58] and certainly had dramatic influence on its biota. Among molluscs, only few species crossed the Apsheronian–Bakunin boundary and major extinction events have been documented for certain groups (e.g. Lymnocardiinae [59]). During this time, a drier and less humid climate persisted [60–62]. In conjunction with the regression, it is not unreasonable to think many of the rivers draining into the Caspian Sea may have become seasonal or dried out for short periods affecting their suitability as refugia. Furthermore, tectonic activity in the Caucasus and Alborz mountains may have affected the lifespan and suitability of springs as refugia [63–66]. Comparable examples of dramatic and sustained low stands have been shown for some ancient lakes to reduce diversity in groups to a single lineage [67]. A well-documented similar case may be Lake Malawi where two recent gastropod species flocks have arisen from a single surviving lineage [68,69].

## 4.2. Phylogenetic relationships and taxonomy

Pontocaspian and southern Iranian *Theodoxus* groups are each monophyletic and probably diverged around 1.60 Ma (95% HPD 2.13–1.7 Ma; figure 3). The age of this split post-dates the older divergences from all other *Theodoxus* species studied by Sands *et al.* [22]. As such, it is highly likely that these *Theodoxus* groups share a sister relationship. On the intraspecific level, however, a strong paraphyly can be detected within each group when comparing the phylogroups and morphospecies included (table 1 and figure 3).

The southern Iranian phylogroups II and III contain specimens attributed to *T. pallidus*, while phylogroups I, III and IV contain specimens attributed to *T. doriae*. This not only shows polyphyly between species on a phylogroup level (phylogroups I, II and IV; table 1 and figure 3), but additionally that there is paraphyly within phylogroups (phylogroup III; table 1 and figure 3). Both species were originally described from Iran and already considered synonymous by Starmühlner & Edlauer [70]. Glöer & Pešić [71] recently reviewed the material of *T. pallidus* studied by Starmühlner & Edlauer [70], but concluded that *T. doriae* should be synonymized with *T. fluviatilis* (Linnaeus, 1758). Our molecular data from topotypic material conforming to Dunker's [72] original description of *T. pallidus* and topotypic material of *T. doriae* from Kerman province support the independence of *T. pallidus* from *T. fluviatilis*, as well as the synonymy of *T. pallidus* and *T. doriae*, as suggested by Starmühlner & Edlauer [70].

Compared with the depth of intraspecific relationships within *T. pallidus*, the intraspecific diversity in Pontocaspian *Theodoxus* is far younger (figure 3). Moreover, more extensive paraphyly and polyphyly of morphospecies is found within and between the two phylogroups. For example, classical *T. major* and *T. pallasi* cluster both within phylogroup V and VI (figure 3). *Theodoxus astrachanicus* is restricted to phylogroup V and *T. schultzii* to phylogroup IV, but there is no support for monophyly of these 'species' (figure 3). *Theodoxus pallasi* and *T. astrachanicus* have been synonymized based on morphological similarities [5,23], which was already supported through previous phylogenetic analyses [22], the inclusion of *T. schultzii* and *T. major* has only been speculated [5]. Our study reaffirms these earlier suggestions and warrants the synonymy of all four morphospecies based on molecular data, with the name *Theodoxus major* Issel, 1865 having priority.

## 4.3. Conservation aspects

Trends observed in the EBSPs for both groups highlight the important aspects for conservation. Following a period of prolonged stability, both the Pontocaspian (*T. major*) and southern Iranian (*T. pallidus*) groups encountered drastic changes in population size, yet in different directions (figure 5). Although a steady population decline in *T. pallidus* beginning around 90 ka needs to be treated with some caution (as a consequence of a potential bias from population structure [45,46]), our balanced sampling strategy may mitigate some of the concern [46]. The population bottleneck observed is additionally corroborated by the TCS haplotype networks and *h* assessments (table 2 and figure 4) and would correlate well with the onset of drier climates and the first appearance of humans in the region [73–75]. While early humans probably had little environmental footprints when compared with today, an ever-growing pressure for water resources in conjunction with these climatic changes [75–78] may have posed (and continue to pose) threats for *T. pallidus*. We advocate for a more intraspecific level EBSP assessment of *T. pallidus* phylogroups to further validate the demographic decline observed. Nevertheless, given that spring systems often contain unique and endemic haplotypes (such as those observed in our own data), and springs themselves are under severe threat in the region [78] dedicated conservation efforts of a variety of spring systems are still critical to protect this species (and some of its unique populations).

Conversely, our analysis suggests that *T. major* has expanded over the last 5000 years (figure 5). Considering the entire range of the 95% confidence interval, however, both a population decline and increase are feasible and are additionally confirmed through the phylogroup-specific EBSPs (electronic supplementary material, figure S1). Already for a number of Pontocaspian species, population declines are evident [5]. Several species of Pontocaspian molluscs are even thought to have gone extinct during the past century [5]. The causes for these declines are not well established, given the general lack of data for most species [5], but may be linked to anthropogenic activity [79]. However, the same causes could have been responsible for the apparent population increase, such as by reducing competition. Consequently, species monitoring and distribution modelling are required to better assess its conservation status.

# 5. Conclusion

Our analyses were able to detect a strong phylogeographic structure in the southern Iranian group, compared with a largely unstructured Pontocaspian *Theodoxus* group. This comparison indicates that the drivers of diversification in Pontocaspian *Theodoxus* are unlikely to be related to the changes in lake level and refugial persistence as in southern Iran. While low lake level and increased salinity cannot be totally excluded from driving intraspecific divergence events in the Pontocaspian region, we found little evidence to suggest low stands have had a significant effect on the phylogeographic structure or demographic history of Pontocaspian *Theodoxus* since the Middle Pleistocene. Rather, high stands and the lack of barriers to gene flow may have caused a homogenizing effect, given the haplotype sharing among specimens from diverse localities. However, it was surprising that all intraspecific diversity diverged from a single lineage during the Middle Pleistocene, which contrasts with the available fossil evidence. We argue that lake level and salinity changes as a consequence of the Tyurkyanian low stand 1.05–0.95 Ma, being one of the longest and most dramatic regressions in the Caspian Sea, coupled with hydrological, climatic and geological changes, may have caused an earlier catastrophic bottleneck event in Pontocaspian *Theodoxus* with only a single lineage persisting. Moreover, our phylogenetic analyses demonstrate extensive paraphyly and polyphyly across several phylogroups of traditionally recognized morphospecies and suggest the presence of only two species; *T. major* and *T. pallidus*, which are endemic to the Pontocaspian region and southern Iran, respectively.

These results not only broaden our understanding of the effects of lake level and salinity changes in the context of Pontocaspian taxa but also provide insights into the evolution and outlooks on the taxonomy and conservation of *Theodoxus*. First, they highlight that the Pontocaspian region may lack geographical barriers to gene flow, even among faunas with limited dispersal capabilities such as *Theodoxus*. Thus, the Caspian Sea may have acted as a single refugium. Second, they suggest Pontocaspian gastropods may be more tolerant to the salinity and lake-level regimes and changes than previously thought. Third, phenotypic variation in *Theodoxus* is obviously high. This could have large implications for regional stratigraphy and environmental reconstruction (which often use molluscs as environmental indicators) as well as invasion biology (as more adaptable species often have higher invasive potential). Finally, species monitoring and modelling in the Pontocaspian region are essential for assessing the need for conservation. Future research would best be directed to investigate the ecological flexibility, distribution and abundance and taxonomy of key Pontocaspian aquatic mollusc species.

Ethics. Nagoya protocol-relevant materials were collected based on ABS agreement NBC-KAPE-270417 with the Kazakhstan Agency of Applied Ecology. Other material derived from Armenia, Azerbaijan, Iran, Russia and Ukraine were sampled through bilateral agreements among collaborating institutes.

Data accessibility. Voucher specimens of all processed material are stored in the Justus Liebig University Giessen, Systematics and Biodiversity collection (UGSB), Germany. DNA sequences were deposited on GenBank, accessions numbers: MN168547–MN168757, MN174926–MN175136 and MN180417–MN180627. All nucleotide sequence alignment files were used as input data for the various analyses, output data used to construct the graphs of the EBSP analyses and db-RDA input files with R script are available through the Dryad Digital Repository: https://dx.doi.org/10.5061/dryad.mn15f80 [80].

Authors' contributions. All authors contributed to the research article. A.F.S. conceptualized the study, A.F.S., S.N., M.F.H. and V.V.A. conducted the fieldwork and organized logistics, A.F.S. and S.N. performed the laboratory work, A.F.S. and T.A.N. analysed the data and A.F.S., T.A.N., T.W. and C.A. led the writing.

Competing interests. The authors declare no competing interests.

Funding. The research herein has received funding from the European Commission's Horizon 2020 research and innovation programme under the grant agreement no. 642973: Drivers of Pontocaspian Biodiversity Rise and Demise (PRIDE). A.F.S. was supported by a Marie Skłodowska-Curie Action Fellowship and T.A.N. by an Alexander von Humboldt scholarship.

Acknowledgements. The authors would like to thank the Institute of Hydroecology and Ichthyology for the Armenian National Academy of Science (Bardukh Gabrielyan), Institute of Zoology for the Azerbaijan National Academy of Sciences (Elman Yusifov and Konul Ahmadova), Kazakhstan Agency for Applied Ecology (Feodor Klimov), Southern Research Center of the Russian Academy of Sciences (Vadim Titov), Laboratory of Macro-Ecology and Biogeography of Invertebrates—St. Petersburg State University (Maxim Vinarski) and Laboratory of Pleistocene Palaeogeography—Moscow State University (Tamara Yanina) for their logistical assistance. Moreover, the authors are grateful especially to Konul Ahmadova, Olga Anistratenko, Shebnem Feteliyeva Farzali, Igor Khaliman, Feodor Klimov, Matteo Lattuada, Sergej Sereda, Maxim Vinarski and Frank Wesselingh for their help in collecting material in the field, Torsten Hauffe for collecting material and analytical assistance and Conrad Matthee for his assistance in the interpretation of some results. Finally, the constructive comments of two anonymous reviewers and the associate editor Kristina Sefc greatly improved the paper.

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
