## [Reviewer comments · Royal Society Open Science]

Review History

RSOS-190965.R0 (Original submission)

Review form: Reviewer 1

Is the manuscript scientifically sound in its present form?

Yes

Are the interpretations and conclusions justified by the results?

No

Is the language acceptable?

Yes

Do you have any ethical concerns with this paper?

No

Have you any concerns about statistical analyses in this paper?

Yes

Recommendation?

Major revision is needed (please make suggestions in comments)

Comments to the Author(s)

This study provides a comparative phylogeographic analyses of *Theodoxus* gastropods from the Pontocaspian region and Southern Iran. The study is based on a good dataset with samples from quite a few distinct localities well spread across the study area. Although the manuscript is generally well written and most of the analyses/conclusions drawn from the analyses are ok, I do have some comments that need to be considered before this ms is ready for publication:

Major comments on analyses:

1. General comment on LTT-plots: LTT plots are typically NOT used for assessing demographic histories, but rather to infer/visualise the accumulation of distinct species/phylogenetic lineages through time. LTTs are not very useful for inferring intraspecific patterns of diversification as here LTTs would only make sense if you had all (or nearly all) the species' haplotypes included. As this is something you cannot be sure of, I strongly recommend removing LTTs from your ms.

2. Bayesian Skyline Plots: You have lots of phylogeographic structure in your data, in particular in the Iranian dataset, but also in the Pontocaspian data (even if you state that there is hardly any population genetic structuring). It is well known that BSPs might be heavily affected by population structure and there are already some studies available that tested the effect of population structure and sampling design on the performance of BSPs. I strongly recommend to check the relevant literature and adapt your analysis strategy and/or interpretation of the BSP patterns accordingly.

Minor comments:

1. Summary & Materials and methods: You write that your study is based on three amplified gene fragments. It's never mentioned you also sequenced them ...

2. I strongly suggest to provide some more information on the biology/ecology of the study species (habitat preference, specialist or generalist, ...) and the hydrological systems(s) (potential habitat discontinuities in the Caspian Sea that are potential barriers to dispersal, hydrological connection to other systems, ...). This info is important for interpreting the patterns you find.

3. Summary, Results & Discussion: Is the resolution (mutation rate) of your markers good enough to test for a potential influence of Pleistocene lake level fluctuations on population structure?

4. Introduction, line 47: How old is the Caspian Sea?

5. Introduction, line 52: In what sense is the Caspian Sea a unique model system to study the effect of palaeo-environmental changes on the phylogeographic structure in long-lived lake biota? It's certainly an excellent model system, but I can think of quite a few other systems that are equally well (or maybe even better) suited - e.g., just think of the East African Great Lakes.

6. M&M, page 2, line 48: Why did you use the BD tree prior? Any specific reasons for doing so? I'm asking because you are looking at a mixture of inter- and (mainly) intraspecific data and tree prior choice might not be a trivial issue here.

7. M&M, page 2, lines 67-68: I guess you mean 95%HPD.
8. M&M, page 3, line 16+: Did you do a BSP (as stated in the ms) or an EPSP analysis? I'm asking because as far as I know the standard BSP analysis does not allow for multilocus data. Please clarify.
8. M&M, page 3, line 25: Did you really use the 95% confidence intervals or rather the 95% HPD intervals (which would be the standard way).
9. Results, page 3, lines 33-34: Three (in Iran) and on (Pontocaspian) divergence events? You cant' say it this way, especially since there is quite some intralineage divergence in the main Pontocaspian haplogroups. Please rephrase.
10. Results, page 3, lines 38+: It would be good to indicate the different morphospecies you refer to here also in the tree.
11. Discussion, page 4, line 1: Where is the evidence for the proposed catastrophic bottleneck? This is not based on your data, or is it? Please clarify.
12. Discussion, page 4, line2 & Conclusions, page 5, line 30: no phylogeographic structure in the Pontocaspian lineage? I disagree here. Judging from the network and Table S1, there's quite some structure, within the three main clades/refugial lineages. I think you can and should put much more emphasis on the phylogeographic patterns in the discussion, e.g. how haplogroup V got distributed over such a large area, from the Black Sea over rivers northwest of the Caspian Sea and the Caspian Sea to rivers southeast of the Caspian Sea, whereas the two subclades of haplogroup VI have very narrow distriptions. By the way, why don't you consider these two subclades different haplogroups? This would make much more sense.
13. Discussion: Please discuss your findings also in light of what is known on habitat preferences of the species. E.g., are Theodoxus habitat specialists or rather generalists? And are there any obvious habitat discontinuities that might act as dispersal barriers?

Review form: Reviewer 2

Is the manuscript scientifically sound in its present form?

Yes

Are the interpretations and conclusions justified by the results?

Yes

Is the language acceptable?

Yes

Do you have any ethical concerns with this paper?

No

Have you any concerns about statistical analyses in this paper?

Yes

Recommendation?

Accept with minor revision (please list in comments)

Comments to the Author(s)

The manuscript reports on a phylogenetic reconstruction of the neritid genus *Theodoxus* from the Caspian Sea region. The authors have a range of samples from across the region, and used three genes (two mitochondrial, one nuclear) for their inferences. There was evidence of a general lack of reciprocal monophyly among species, but instead isolation by distance was present. The authors also undertook reconstructions of historic population sizes using their data, and speculate on potential links between climate and present day diversity.

Overall this is a nice paper based on a hard-won dataset. The analyses appear appropriate to the questions being asked, and the results are fairly clear. Most of my comments relate to relatively minor issues of presentation.

P2, L40-43. I'm not convinced there is any evidence here of effects of low lake levels on population genetic structure. At best the evidence is that high stands may have provided opportunities for genetic homogenisation.

P2, L58. use "only a small number"

P3, L23. Use "should possess similar"

P3, L35. I'm not convinced you give any insights into plasticity here (it doesn't appear to be mentioned elsewhere in the paper). Indeed, these data alone do not tell us anything about the plasticity, or otherwise, of these phenotypes.

P3, L41. I am not familiar with the tests you used for saturation (it would be helpful if you named and cited the specific test used), but typically a $p < 0.05$ is considered statistically significant.

P3, L46. You say taxa, but this is a pleural of taxon (a group of one or more populations of an organism). Surely you mean individual.

P3, L~60. 50% burn-in removed?

P4, L3. No further burn-in removed.

P4, L32. It is not clear how your "phylogroups" were delimited.

P5, L1. I'm not sure you found any evidence for a catastrophic bottleneck. The data are, I feel, being overinterpreted.

P6, L6. RE synonymy. Your analysis is based on effectively 2 independent loci (both mtDNA genes are linked). Could such patterns arise from parallel evolution?

P6, L35. "contrasts with the fossil evidence"

P9, L37. Zookeys volume absent.

P11, Table. All your p-values for your isolation by distance tests are identical? It is not clear to me how you tested for isolation by distance, but since Mantel tests are now suggested to be flawed you should consider distance-based redundancy analysis in the vegan R package.

P13-15, Figures. Your colours are indistinguishable. I suggest you reconsider these, perhaps use letter and colour combinations.

P17, L37. What is "intraspecific diversification" in this sense? Why should it only start at this point? It doesn't make any logical sense to me.

Decision letter (RSOS-190965.R0)

28-Jun-2019

Dear Mr Sands,

The editors assigned to your paper ("Old lake vs. young taxa: a comparative phylogeographic perspective on the evolution of Caspian Sea gastropods (Neritidae: *Theodoxus*)") have now received comments from reviewers. We would like you to revise your paper in accordance with the referee and Associate Editor suggestions which can be found below (not including

confidential reports to the Editor). Please note this decision does not guarantee eventual acceptance.

Please submit a copy of your revised paper before 21-Jul-2019. Please note that the revision deadline will expire at 00.00am on this date. If we do not hear from you within this time then it will be assumed that the paper has been withdrawn. In exceptional circumstances, extensions may be possible if agreed with the Editorial Office in advance. We do not allow multiple rounds of revision so we urge you to make every effort to fully address all of the comments at this stage. If deemed necessary by the Editors, your manuscript will be sent back to one or more of the original reviewers for assessment. If the original reviewers are not available, we may invite new reviewers.

- Data accessibility

<http://datadryad.org/submit?journalID=RSOS&manu=RSOS-190965>

- Competing interests

- Authors' contributions

- Acknowledgements

- Funding statement

Kind regards,
Lianne Parkhouse
Editorial Coordinator
Royal Society Open Science
openscience@royalsociety.org

on behalf of Dr Kristina Sefc (Associate Editor) and Kevin Padian (Subject Editor)
openscience@royalsociety.org

Subject Editor's comments (Professor Kevin Padian):

Thanks for your submission. We hope you find the reviewers' comments constructive as you revise, and we look forward to the next version. Best wishes.

Associate Editor's comments (Dr Kristina Sefc):

The reviewers of the manuscript are both impressed by the remarkable phylogeographic sampling underlying the study, and are mostly in agreement with the presented work. Please respond to the concerns (as raised by reviewer 1) regarding the use of lineage-through-time plots and Bayesian skyline plots in your analysis. I agree with the reviewer that it is preferable to omit analyses if assumptions are violated or if there's a risk to arrive at spurious results. Regarding the BSP, if a literature survey (e.g. simulation studies testing effects of population structure on BSP analyses) suggests that the structure in your data might pose a problem, you could consider to

run analyses for the individual haplogroups separately and omit divergent lineages that include only few haplotypes.

Please also provide the additional information and discussions as suggested by reviewer 1, and follow the comments of reviewer 2.

Reviewers' Comments to Author:

Reviewer: 1

This study provides a comparative phylogeographic analyses of *Theodoxus* gastropods from the Pontocaspian region and Southern Iran. The study is based on a good dataset with samples from quite a few distinct localities well spread across the study area. Although the manuscript is generally well written and most of the analyses/conclusions drawn from the analyses are ok, I do have some comments that need to be considered before this ms is ready for publication:

Major comments on analyses:

1. General comment on LTT-plots: LTT plots are typically NOT used for assessing demographic histories, but rather to infer/visualise the accumulation of distinct species/ phylogenetic lineages through time. LTTs are not very useful for inferring intraspecific patterns of diversification as here LTTs would only make sense if you had all (or nearly all) the species' haplotypes included. As this is something you cannot be sure of, I strongly recommend removing LTTs from your ms.

2. Bayesian Skyline Plots: You have lots of phylogeographic structure in your data, in particular in the Iranian dataset, but also in the Pontocaspian data (even if you state that there is hardly any population genetic structuring). It is well known that BSPs might be heavily affected by population structure and there are already some studies available that tested the effect of population structure and sampling design on the performance of BSPs. I strongly recommend to check the relevant literature and adapt your analysis strategy and/or interpretation of the BSP patterns accordingly.

Minor comments:

1. Summary & Materials and methods: You write that your study is based on three amplified gene fragments. It's never mentioned you also sequenced them ...

2. I strongly suggest to provide some more information on the biology/ecology of the study species (habitat preference, specialist or generalist, ...) and the hydrological systems(s) (potential habitat discontinuities in the Caspian Sea that are potential barriers to dispersal, hydrological connection to other systems, ...). This info is important for interpreting the patterns you find.

3. Summary, Results & Discussion: Is the resolution (mutation rate) of your markers good enough to test for a potential influence of Pleistocene lake level fluctuations on population structure?

4. Introduction, line 47: How old is the Caspian Sea?

5. Introduction, line 52: In what sense is the Caspian Sea a unique model system to study the effect of palaeo-environmental changes on the phylogeographic structure in long-lived lake biota? It's certainly an excellent model system, but I can think of quite a few other systems that are equally well (or maybe even better) suited - e.g., just think of the East African Great Lakes.

6. M&M, page 2, line 48: Why did you use the BD tree prior? Any specific reasons for doing so? I'm asking because you are looking at a mixture of inter- and (mainly) intraspecific data and tree prior choice might not be a trivial issue here.
7. M&M, page 2, lines 67-68: I guess you mean 95%HPD.
8. M&M, page 3, line 16+: Did you do a BSP (as stated in the ms) or an EPSP analysis? I'm asking because as far as I know the standard BSP analysis does not allow for multilocus data. Please clarify.
8. M&M, page 3, line 25: Did you really use the 95% confidence intervals or rather the 95% HPD intervals (which would be the standard way).
9. Results, page 3, lines 33-34: Three (in Iran) and on (Pontocaspian) divergence events? You can't say it this way, especially since there is quite some intralinesage divergence in the main Pontocaspian haplogroups. Please rephrase.
10. Results, page 3, lines 38+: It would be good to indicate the different morphospecies you refer to here also in the tree.
11. Discussion, page 4, line 1: Where is the evidence for the proposed catastrophic bottleneck? This is not based on your data, or is it? Please clarify.
12. Discussion, page 4, line 2 & Conclusions, page 5, line 30: no phylogeographic structure in the Pontocaspian lineage? I disagree here. Judging from the network and Table S1, there's quite some structure, within the three main clades/refugial lineages. I think you can and should put much more emphasis on the phylogeographic patterns in the discussion, e.g. how haplogroup V got distributed over such a large area, from the Black Sea over rivers northwest of the Caspian Sea and the Caspian Sea to rivers southeast of the Caspian Sea, whereas the two subclades of haplogroup VI have very narrow distributions. By the way, why don't you consider these two subclades different haplogroups? This would make much more sense.
13. Discussion: Please discuss your findings also in light of what is known on habitat preferences of the species. E.g., are *Theodoxus* habitat specialists or rather generalists? And are there any obvious habitat discontinuities that might act as dispersal barriers?

Reviewer: 2

The manuscript reports on a phylogenetic reconstruction of the neritid genus *Theodoxus* from the Caspian Sea region. The authors have a range of samples from across the region, and used three genes (two mitochondrial, one nuclear) for their inferences. There was evidence of a general lack of reciprocal monophyly among species, but instead isolation by distance was present. The authors also undertook reconstructions of historic population sizes using their data, and speculate on potential links between climate and present day diversity.

Overall this is a nice paper based on a hard-won dataset. The analyses appear appropriate to the questions being asked, and the results are fairly clear. Most of my comments relate to relatively minor issues of presentation.

P2, L40-43. I'm not convinced there is any evidence here of effects of low lake levels on population genetic structure. At best the evidence is that high stands may have provided opportunities for genetic homogenisation.

P2, L58. use “only a small number”

P3, L23. Use “should possess similar”

P3, L35. I’m not convinced you give any insights into plasticity here (it doesn’t appear to be mentioned elsewhere in the paper). Indeed, these data alone do not tell us anything about the plasticity, or otherwise, of these phenotypes.

P3, L41. I am not familiar with the tests you used for saturation (it would be helpful if you named and cited the specific test used), but typically a $p < 0.05$ is considered statistically significant.

P3, L46. You say taxa, but this is a plural of taxon (a group of one or more populations of an organism). Surely you mean individual.

P3, L~60. 50% burn-in removed?

P4, L3. No further burn-in removed.

P4, L32. It is not clear how your “phylogroups” were delimited.

P5, L1. I’m not sure you found any evidence for a catastrophic bottleneck. The data are, I feel, being overinterpreted.

P6, L6. RE synonymy. Your analysis is based on effectively 2 independent loci (both mtDNA genes are linked). Could such patterns arise from parallel evolution?

P6, L35. “contrasts with the fossil evidence”

P9, L37. Zookeys volume absent.

P11, Table. All your p-values for your isolation by distance tests are identical? It is not clear to me how you tested for isolation by distance, but since Mantel tests are now suggested to be flawed you should consider distance-based redundancy analysis in the vegan R package.

P13-15, Figures. Your colours are indistinguishable. I suggest you reconsider these, perhaps use letter and colour combinations.

P17, L37. What is “intraspecific diversification” in this sense? Why should it only start at this point? It doesn’t make any logical sense to me.

Author's Response to Decision Letter for (RSOS-190965.R0)

See Appendix A.

RSOS-190965.R1 (Revision)

Review form: Reviewer 1

Is the manuscript scientifically sound in its present form?

Yes

Are the interpretations and conclusions justified by the results?

No

Is the language acceptable?

Yes

Do you have any ethical concerns with this paper?

No

Have you any concerns about statistical analyses in this paper?

Yes

Recommendation?

Accept with minor revision (please list in comments)

Comments to the Author(s)

Most of my previous concerns have been addressed satisfactorily. Two issues, however, remain:

1. lines 175-177 (and comment 10 of the original review): ... no, I didn't misread the sentence. Note that each node in the tree represents a divergence event. In fact, you can delete this entire sentence as it essentially conveys the same information as the previous sentence (lines 174-175).

2. EBSPs: Yes, doing separate analyses for the two Pontocaspian phylogroups (now presented as Supplementary info) is the right way to deal with these data. And yes, for most of the Iranian phylogroups sample size/diversity is too small to allow for doing EBSP analyses. This, however, is no justification for pooling all these four pretty divergent phylogroups (cryptic species?) for a single EBSP analysis. You have virtually no intraphylogroup diversity in the Iranian samples and with the large interphylogroup divergence you inevitably get a signature of a drastic recent population decline in your EBSP. It's simply not possible to do reliable EBSP analyses with these Iranian samples. I suggest you have a look at previous studies that evaluated the performance of (E)BSPs in the presence of population structure. Specifically, in highly structured populations, drastic recent population size declines were observed (similar to what you found with your data) even though the sequences were simulated under a constant population size scenario. Two relevant studies that come to my mind are Heller et al. 2013 PLoS One & Grant 2015 J Hered. I think you'll have to have a more critical look at your EBSPs, especially since these data are essential for your discussion/conclusions on conservation aspects in your gastropods.

Decision letter (RSOS-190965.R1)

27-Aug-2019

Dear Mr Sands:

Manuscript ID RSOS-190965.R1 entitled "Old lake vs. young taxa: a comparative phylogeographic perspective on the evolution of Caspian Sea gastropods (Neritidae: *Theodoxus*)" which you submitted to Royal Society Open Science, has been reviewed. The comments of the reviewer(s) are included at the bottom of this letter.

Please submit a copy of your revised paper before 19-Sep-2019. Please note that the revision deadline will expire at 00.00am on this date. If we do not hear from you within this time then it will be assumed that the paper has been withdrawn. In exceptional circumstances, extensions may be possible if agreed with the Editorial Office in advance. We do not allow multiple rounds of revision so we urge you to make every effort to fully address all of the comments at this stage. If deemed necessary by the Editors, your manuscript will be sent back to one or more of the original reviewers for assessment. If the original reviewers are not available we may invite new reviewers.

To revise your manuscript, log into <http://mc.manuscriptcentral.com/rsos> and enter your

Author Centre, where you will find your manuscript title listed under "Manuscripts with Decisions." Under "Actions," click on "Create a Revision." Your manuscript number has been appended to denote a revision. Revise your manuscript and upload a new version through your Author Centre.

- Ethics statement

- Data accessibility

- Competing interests

- Authors' contributions

- Acknowledgements

- Funding statement

Kind regards,

Alice Power

Editorial Coordinator

on behalf of Dr Kristina Sefc (Associate Editor) and Kevin Padian (Subject Editor)

Associate Editor Comments to Author (Dr Kristina Sefc):

Dear authors,

I'd like to thank you for addressing most of the concerns raised in the first round of review. The issue of pooling divergent lineages for the EBSPs still remains and actually represents a rather serious one since an important conclusion - regarding the decline in population size - may be based on an artifact in the analysis. Please see the reviewer's comments for details. If the sampling / the data don't allow to analyse the demographic history of the Iranian populations, then you'll have to consider dropping this part from the manuscript (or interpret the network structures of the mtDNA and ncDNA sequences verbally - less diversity in the mt than nc genomes within each phylogroup may indeed point to a recent bottleneck); or at least discuss the problem associated with pooling the divergent lineages and make clear that there's a risk of a spurious result.

Sincerely, Kristina Sefc

Subject Editor Comments to Authors:

Thank you for addressing previous comments. Please note that the reviewer and the AE still feel there is a major issue to address. In your resubmission please make the necessary edits and address the comments. If the AE does not feel these are sufficiently addressed we will not be able to consider the manuscript further. Best wishes.

Reviewer comments to Author:

Reviewer: 1

Most of my previous concerns have been addressed satisfactorily. Two issues, however, remain:

1. lines 175-177 (and comment 10 of the original review): ... no, I didn't misread the sentence. Note that each node in the tree represents a divergence event. In fact, you can delete this entire sentence as it essentially conveys the same information as the previous sentence (lines 174-175).

2. EBSPs: Yes, doing separate analyses for the two Pontocaspian phylogroups (now presented as

Supplementary info) is the right way to deal with these data. And yes, for most of the Iranian phylogroups sample size/diversity is too small to allow for doing EBSP analyses. This, however, is no justification for pooling all these four pretty divergent phylogroups (cryptic species?) for a single EBSP analysis. You have virtually no intraphylogroup diversity in the Iranian samples and with the large interphylogroup divergence you inevitably get a signature of a drastic recent population decline in your EBSP. It's simply not possible to do reliable EBSP analyses with these Iranian samples. I suggest you have a look at previous studies that evaluated the performance of (E)BSPs in the presence of population structure. Specifically, in highly structured populations, drastic recent population size declines were observed (similar to what you found with your data) even though the sequences were simulated under a constant population size scenario. Two relevant studies that come to my mind are Heller et al. 2013 PLoS One & Grant 2015 J Hered. I think you'll have to have a more critical look at your EBSPs, especially since these data are essential for your discussion/conclusions on conservation aspects in your gastropods.

Author's Response to Decision Letter for (RSOS-190965.R1)

See Appendix B.

Decision letter (RSOS-190965.R2)

18-Sep-2019

Dear Mr Sands,

I am pleased to inform you that your manuscript entitled "Old lake vs. young taxa: a comparative phylogeographic perspective on the evolution of Caspian Sea gastropods (Neritidae: *Theodoxus*)" is now accepted for publication in Royal Society Open Science.

Kind regards,
Lianne Parkhouse
Royal Society Open Science
openscience@royalsociety.org

on behalf of Dr Kristina Sefc (Associate Editor) and Kevin Padian (Subject Editor)
openscience@royalsociety.org

Comments to the Author:

I'd like to thank the authors for the revised discussion of the results of the demographic analysis, which now include reference to potential methodological caveats.

Appendix A

To the Chief Editor, Royal Society Open Science

Dear Prof. Jeremy Sanders CBE FRS

Cc: Prof. Kristina Sefc (Associate Editor), Prof. Kevin Padian (Subject Editor)

Reference Number: RSOS-190965

We very much appreciate the time and effort of the reviewers and the editors in assessing our paper and we are grateful for their useful comments. Below we list all critical comments and suggestions of the reviewers (in italics), together with our point-by-point replies (in blue). Attached at the very end is the revised version of the research paper with the 'Track Changes – All Markup' enabled, for easy reference of where changes have been made.

ASSOCIATE EDITOR (Dr Kristina Sefc):

Please respond to the concerns (as raised by reviewer 1) regarding the use of lineage-through-time plots and Bayesian skyline plots in your analysis. I agree with the reviewer that it is preferable to omit analyses if assumptions are violated or if there's a risk to arrive at spurious results. Regarding the BSP, if a literature survey (e.g. simulation studies testing effects of population structure on BSP analyses) suggests that the structure in your data might pose a problem, you could consider to run analyses for the individual haplogroups separately and omit divergent lineages that include only few haplotypes.

We have removed the LTT plots and improved our Bayesian skyline analyses as suggested. Please see replies to reviewer 1: major comments 1 & 2 and minor comment 8.

REVIEWER: 1

MAJOR COMMENTS

- 1) General comment on LTT-plots: LTT plots are typically NOT used for assessing demographic histories, but rather to infer/visualise the accumulation of distinct species/phylogenetic lineages through time. LTTs are not very useful for inferring intraspecific patterns of diversification as here LTTs would only make sense if you had all (or nearly all) the species' haplotypes included. As this is something you cannot be sure of, I strongly recommend removing LTTs from your ms.*

We agree with the suggestion and have remove the LTT plots and all content related to them.

- 2) Bayesian Skyline Plots: You have lots of phylogeographic structure in your data, in particular in the Iranian dataset, but also in the Pontocaspian data (even if you state that there is hardly and population genetic structuring). It is*

well known that BSPs might be heavily affected by population structure and there are already some studies available that tested the effect of population structure and sampling design on the performance of BSPs. I strongly recommend to check the relevant literature and adapt your analysis strategy and/or interpretation of the BSP patterns accordingly.

We are interested in the overall trends in both groups rather than the phylogroup specific trends. As most phylogroups have overlapping ranges (and thus likely have experienced similar environmental histories), the impact bias of a specific phylogroup on the overall pattern should be limited. Moreover, some phylogroups (specifically those in Iran) are too small to construct reliable skyline plots individually. Nevertheless we note the reviewer's concern and the associate editor's suggestions and have constructed EBS plots (also see minor comment 8 of reviewer 1) for all phylogroups where ESS values < 200 could be established (specifically phylogroups V and VI). These show little deviation from the overall trend from the Pontocaspian group. We have added the new additional results to the supplementary information and made alterations to the methods and results chapters. See: L150–168 & L199–208 and Appendix A; figure S1.

MINOR COMMENTS

- 1) *Summary & Materials and methods: You write that your study is based on three amplified gene fragments. It's never mentioned you also sequenced them ...*

We have now added sequencing information in the 'materials and methods' section. See: L100-101.

- 2) *I strongly suggest to provide some more information on the biology/ecology of the study species (habitat preference, specialist or generalist, ...) and the hydrological systems(s) (potential habitat discontinuities in the Caspian Sea that are potential barriers to dispersal, hydrological connection to other systems, ...). This info is important for interpreting the patterns you find.*

Added as suggested. See: L65–68.

- 3) *Summary, Results & Discussion: Is the resolution (mutation rate) of your markers good enough to test for a potential influence of Pleistocene lake level fluctuations on population structure?*

This largely depends on the duration of the lake level fluctuation and is critically discussed (see L222–224 & L237–239). Moreover, there are a number of recently published studies on *Theodoxus*, using these three genes, to assess phylogeographic patterns and evolutionary history on the intraspecific level with success (i.e. showed to have sufficient resolution to detect a variety of major influences) [1–5]. As most *Theodoxus* intraspecific diversification occurs after the

Pliocene [6], this suggest they variable enough to detect the effect of major lake level changes during this period.

4) *Introduction, line 47: How old is the Caspian Sea?*

Added the age of the Caspian Sea as suggested. See: L37–38.

5) *Introduction, line 52: In what sense is the Caspian Sea a unique model system to study the effect of palaeo-environmental changes on the phylogeographic structure in long-lived lake biota? It's certainly an excellent model system, but I can think of quite a few other systems that are equally well (or maybe even better) suited - e.g., just think of the East African Great Lakes.*

We have changed 'unique', to 'suitable'. See: L44.

6) *M&M, page 2, line 48: Why did you use the BD tree prior? Any specific reasons for doing so? I'm asking because you are looking at a mixture of inter- and (mainly) intraspecific data and tree prior choice might not be a trivial issue here.*

While the 'coalescent population (constant and exponential) tree priors' are more orientated towards population level studies, we selected the birth death-prior due to having more than one species and not wanting to assume all Caspian Sea morphospecies belonged to the same phylogenetic species. Moreover, according to the latest published research [7], the choice of tree prior and molecular clock does not substantially affect phylogenetic patterns or diversification rates.

7) *M&M, page 2, lines 67-68: I guess you mean 95%HPD.*

Changed as suggested. Added '95%' in front of HPD. See: L123–124.

8) *M&M, page 3, line 16+: Did you do a BSP (as stated in the ms) or an EBSP analysis? I'm asking because as far as I know the standard BSP analysis does not allow for multi locus data. Please clarify.*

Changed as suggested. All BS plots have now been reconstructed using the EBSP method and content/references adapted. See: L26, L150–168, L199–208 & L298–307, figure 5 + caption and Appendix A; figure S1.

9) *M&M, page 3, line 25: Did you really use the 95% confidence intervals or rather the 95% HPD intervals (which would be the standard way).*

This should have been 'HPD'. Changed as suggested. See: L120.

10) *Results, page 3, lines 33-34: Three (in Iran) and one (Pontocaspian) divergence events? You can't say it this way, especially since there is quite some intralineage divergence in the main Pontocaspian haplogroups. Please rephrase.*

We feel the reviewer may have misread the sentence (See L174–176). We specifically state 'between phylogroups' and not 'within and between phylogroups'. We are specifically describing the divergence events of I, II, III and IV from each other; and V and VI from each other.

11) *Results, page 3, lines 38+: It would be good to indicate the different morphospecies you refer to here also in the tree.*

Changed as suggested. We have adapted figure 3 to also indicate species names. See: figure 3 + caption.

12) *Discussion, page 4, line 1: Where is the evidence for the proposed catastrophic bottleneck? This is not based on your data, or is it? Please clarify.*

We partly agree with the reviewer, in that the early bottleneck discussed is somewhat speculative as the extended Bayesian skyline plots cannot cover this period of time to adequately test it. The reasoning is based on the presence of diverse fossil data in the absence of early molecular diversification in the phylogeny (all diversity stems from a single lineage over this period) and low nucleotide diversity (indicating a young species flock). This is a reasonable explanation given the data available and the results we have. However we do critically discuss our interpretation and note the speculativeness thereof. See: L212, L249, L256 & L327–328.

13) *Discussion, page 4, line2 & Conclusions, page 5, line 30: no phylogeographic structure in the Pontocaspian lineage? I disagree here. Judging from the network and Table S1, there's quite some structure, within the three main clades/refugial lineages. I think you can and should put much more emphasis on the phylogeographic patterns in the discussion, e.g. how haplogroup V got distributed over such a large area, from the Black Sea over rivers northwest of the Caspian Sea and the Caspian Sea to rivers southeast of the Caspian Sea, whereas the two subclades of haplogroup VI have very narrow distributions. By the way, why don't you consider these two subclades different haplogroups? This would make much more sense.*

We agree with the reviewer that more discussion needs to be included regarding the broad distribution of phylogroups V and VI. As such we have added some discussion about the effects of high stands and how they may have driven homogeneity of

haplotypes across the system (also see response to reviewer 2, minor comment 1). However the two 'subclades' in phylogroup VI are not supported by posterior probabilities in the phylogeny and nor are the differences conserved across the AMOVAs of all genes. As such, discussion of 'narrow ranges' or the general patterns within subclades of phylogroup V or VI would be very dubious.

14) *Discussion: Please discuss your findings also in light of what is known on habitat preferences of the species. E.g., are Theodoxus habitat specialists or rather generalists? And are there any obvious habitat discontinuities that might act as dispersal barriers?*

We have added discussion on this topic. See: L241–247.

REVIEWER: 2

MINOR COMMENTS

1) *P2, L40-43. I'm not convinced there is any evidence here of effects of low lake levels on population genetic structure. At best the evidence is that high stands may have provided opportunities for genetic homogenisation.*

We agree with the reviewer that there is little evidence to support low stands having any effect on population structure; indeed that is primarily our outcome. However, following the suggestion, we have adapted our text to put more focus on the homogenising effects of high stands. See: L30–31, L240–247 & L322–324.

2) *P2, L58. use "only a small number"*

Changed 'low' to 'small' as suggested. See: L51.

3) *P3, L23. Use "should possess similar"*

Changed 'mimic' to 'possess' as suggested. See: L85.

4) *P3, L35. I'm not convinced you give any insights into plasticity here (it doesn't appear to be mentioned elsewhere in the paper). Indeed, these data alone do not tell us anything about the plasticity, or otherwise, of these phenotypes.*

We agree. We have removed the sub-aim to 'provide perspective on phenotypic plasticity'. See: L86.

- 5) P3, L41. *I am not familiar with the tests you used for saturation (it would be helpful if you named and cited the specific test used), but typically a $p < 0.05$ is considered statistically significant.*

There is no specific name for the test for saturation, however we have added in an additional reference as requested. Regarding the level of significance: we gave the p values as found by the analyses (i.e. $p < 0.001$) rather than the level of significance. See: L103–105.

- 6) P3, L46. *You say taxa, but this is a plural of taxon (a group of one or more populations of an organism). Surely you mean individual.*

Changed 'taxa' to 'individuals' as suggested. See: L110.

- 7) P3, L~60. *50% burn-in 'removed'?*

Added 'removed' as suggested. See: L128.

- 8) P4, L3. *No further burn-in 'removed'.*

Added 'removed' as suggested. See: L131.

- 9) P4, L32. *It is not clear how your "phylogroups" were delimited.*

We have now included a small explanation to the methods of how the phylogroups were delimited. See: L131–132.

- 10)P5, L1. *I'm not sure you found any evidence for a catastrophic bottleneck. The data are, I feel, being over interpreted.*

See reply to reviewer 1, minor comment 12.

- 11)P6, L6. RE synonymy. Your analysis is based on effectively 2 independent loci (both mtDNA genes are linked). Could such patterns arise from parallel evolution?

Parallel evolution is unlikely given its rarity, especially considering that the amount of mutations that have occurred and the extended geological time over which these took place. Moreover, although not discussed or shown, the majority of key morphological characters are conserved, such as those related to the operculum (which is often used to distinguish species). As this is not really in the scope of this study, it will be addressed in a follow-up article currently in preparation by the authors reviewing the taxonomy and morphology of all extant *Theodoxus* spp.

12)P6, L35. “contrasts with the fossil evidence”

Added 'with' as suggested. See: L325

13)P9, L37. Zookeys volume absent.

Volume number now added. See: reference 70.

14)P11, Table. All your p-values for your isolation by distance tests are identical? It is not clear to me how you tested for isolation by distance, but since Mantel tests are now suggested to be flawed you should consider distance-based redundancy analysis in the vegan R package.

We have incorporated the distance-based redundancy analysis in R as suggested and adjusted the text accordingly. See: L25, L142–147 & L194–196, table 2 and Appendix B.

15)P13-15, Figures. Your colours are indistinguishable. I suggest you reconsider these, perhaps use letter and colour combinations.

Changed as suggested. See: figures 2-4.

16)P17, L37. What is “intraspecific diversification” in this sense? Why should it only start at this point? It doesn't make any logical sense to me.

'Intraspecific diversification' refers here to the date at which the skyline plots begin (i.e. the starting point of the observed diversification in the group). We have adapted the caption of figure 5 to make this clear.

REFERENCE LIST FOR REPLY COMMENTS

1. Bunje PME. 2007 Fluvial range expansion, allopatry, and parallel evolution in a Danubian snail lineage (Neritidae: *Theodoxus*). *Biol. J. Linn. Soc.* **90**, 603–617. (doi:10.1111/j.1095-8312.2007.00750.x)
2. Bunje PME. 2005 Pan-European phylogeography of the aquatic snail *Theodoxus fluviatilis* (Gastropoda: Neritidae). *Mol. Ecol.* **14**, 4323–4340. (doi:10.1111/j.1365-294X.2005.02703.x)
3. Fehér Z, Zettler ML, Bozsó M, Szabó K. 2009 An attempt to reveal the systematic relationship between *Theodoxus prevostianus* (C. Pfeiffer, 1828) and *Theodoxus danubialis* (C. Pfeiffer, 1828) (Mollusca, Gastropoda, Neritidae). *Mollusca* **27**, 95–107.
4. Fehér Z, Albrecht C, Major Á, Sereda S V., Krízsik V. 2012 Extremely low genetic diversity in the endangered striped nerite, *Theodoxus transversalis*

(Mollusca, Gastropoda, Neritidae) – a result of ancestral or recent effects? *North. West. J. Zool.* **8**, 300–307.

5. Gergs R, Koester M, Grabow K, Schöll F, Thielsch A, Martens A. 2015 *Theodoxus fluviatilis*' re-establishment in the River Rhine: A native relict or a cryptic invader? *Conserv. Genet.* **16**, 247–251. (doi:10.1007/s10592-014-0651-7)
6. Sands AF, Sereda SV, Stelbrink B, Neubauer TA, Lazarev S, Wilke T, Albrecht C. 2019 Contributions of biogeographical functions to species accumulation may change over time in refugial regions. *J. Biogeogr.* **46**, 1274–1286.
7. Sarver BAJ, Pennell MW, Brown JW, Keeble S, Hardwick KM, Sullivan J, Harmon LJ. 2019 The choice of tree prior and molecular clock does not substantially affect phylogenetic inferences of diversification rates. *PeerJ* **7**, e6334. (doi:10.7717/peerj.6334)

[revised manuscript text omitted]

We hypothesise that repeated ~~fluctuations in lake level and Caspian Sea low stands with increased~~ salinity (~~perhaps~~
~~reflected in the phenotypic variability; figure 1~~) resulted in multiple isolated refugia across the Caspian Sea and its
catchments; and, thus a high degree of population structure. As such, Pontocaspian and Iranian *Theodoxus* should
~~mimic~~ possess similar population structures. Additionally, as the taxonomy of some of these species is poorly
resolved, we provide a molecular perspective on species identities ~~and phenotypic plasticity~~. The outcomes of this
research should identify the timing and extent of how major salinity and lake level changes may have affected the
evolutionary histories of native Pontocaspian aquatic gastropods and identify Pontocaspian refugia (particularly
across the Caspian Sea drainage network). Moreover, we discuss how our results may help resolve taxonomic
uncertainties and what the implications may mean for the conservation of Pontocaspian taxa.

3. Materials and Methods

3.1 Sample collection and laboratory protocols

*Theodoxus* specimens were collected and stored following Sands et al. [6]. We included the genetic data of either five
or ten specimens per location to allow for robust analyses (table 1; figure 2). Sands' et al. [6] protocols were again
followed to extract and amplify two mtDNA fragments; cytochrome c oxidase subunit I (COI) and 16S rRNA (16S_r)
and one nDNA intron fragment, ATP synthetase subunit alpha (ATP α). Total genomic DNA was extracted from foot
tissue of the snails using a DNeasy Blood and Tissue kit (QIAGEN, Hilden, Germany) and amplified using the
primers TheoSF1 and TheoSR1 for COI [6], 16Sar-L and 16Sbr-H for 16S [30]; and ATP α Saf1 and ATP α Sar1 for
ATP α [31] – for primer sequences and PCR conditions see Sands et al. [6]. Purification and bidirectional Sanger
sequencing of the amplified gene fragments were carried out by LGC Ltd. (Berlin, Germany). Where needed, we
incorporated published GenBank sequences of individuals where all three DNA fragments were available. Sequence
ends were trimmed and aligned in Genious 10.1.2 [32] using the Genious alignment algorithm. To test for saturation
over 1,000 replicates [33], we used DAMBE 5.0.52 [34,35]. None of the three molecular datasets showed any
significant degree of saturation (COI, 16S and ATP α ; $p < 0.001$).

3.2 Phylogenetic analyses

To investigate the relationships among all *Theodoxus* specimens, delimit phylogroups and provide divergence date
estimates between and within the Pontocaspian and southern Iranian groups, a dated Bayesian phylogeny was
constructed. The phylogeny combined all amplified gene fragments (1,609 base pairs in total) for 222 taxa individuals
(211 sequenced for the first time; table 1).

First, a log-normal relaxed clock was set for all gene datasets and a birth-death tree prior was selected. bModelTest
v1.1.2 [36], as implemented in BEAST v2.5.2 [37], was used to determine the best-fit model for each gene set. TN93
(121131) was determined the optimal model for COI and ATP α ; and a variant of the HKY (with an additional group
for the rates r_{AT} and r_{AG} ; 121323) for 16S. Second, sequences derived from *T. jordani* (Sowerby, 1836) and the more
distantly related *T. transversalis* (Pfeiffer, 1828) were used to root the phylogeny (see [6]). As fossil dating is
challenging for *Theodoxus*, the phylogeny was dated calibrated using the molecular clock rates and secondary dating
of internal nodes as established by Sands et al. [6]. For congruency, clock rates and secondary dates were set to the

95% confidence intervals and highest posterior densities (HPD) established therein. Molecular clock rate priors were
linearly distributed (COI, 1.78%, standard deviation (SD) 2.18%–1.40%; 16S, 0.73%, SD 0.98%–0.48%; ATP α ,
0.65%, SD 0.95%–0.37%), while secondary dating priors were set to normal distribution (*T. transversalis* from all
specimens, 8.27 million years ago (Ma), highest posterior density (HPD) 11.60–4.90 Ma; *T. jordani* from all
in-group specimens, 4.77 Ma, 95% HPD 6.33–3.20 Ma).

Thereafter, two independent runs of 200,000,000 MCMC generations, saving one tree in every 20,000 generations,
were constructed in BEAUti v2.5.2 [37] and implemented in BEAST v2.5.2 through the CIPRES Science Gateway
[38]. LogCombiner v2.5.2 [37] was used to combine trees and log files of each run with 50% burn-in removed.
Validation of convergence and mixing of the combined log file was assessed in Tracer v1.7.1 [39] to ensure that all
effective sample size (ESS) values were > 200, and TreeAnnotator v2.5.2 [37] was used to summarize trees, with no
further burn-in removed. Phylogroups were delimited based on posterior probability (PP) support values, where the
most recent daughter lineages stemming from a common ancestor were both significantly supported (PP > 0.95).

3.3 Phylogeographic structure

Evolutionary relationships among COI, 16S and ATP α haplotypes of the Pontocaspian and southern Iranian
*Theodoxus* groups were established independently for each group and gene through statistical haplotype networks in
TCS v1.21 [40]. Sequence ambiguities associated with heterozygote states in the ATP α sequences (nDNA) were
resolved by determining 90% probability score alleles in Phase v2.1 [41] as implemented in DnaSP v6.11.01 [42]
under default settings. To further test for differentiation within each group, among geographical sampling localities,
analyses of molecular variance (AMOVAs) were performed in Arlequin v3.5.2.2 [43]. P-values were subjected to
Holm's sequential Bonferroni corrections [44]. Furthermore, Arlequin was used to calculate haplotypic (h) and
nucleotide diversities (π) and finally, isolation by distance (IBD) was tested for using the distance-based
redundancy analysis (db-RDA) [45,46] with the package *vegan* v2.5-4 [47] in the R statistical environment v3.5.2
[48]. Since longitude, latitude and geographic distance among localities were strongly correlated, we only included
the latter parameter in the analyses. Principal coordinates analyses was performed on the matrices of pairwise
geographic distances and the first principal component was used as single continuous variable in the db-RDA [49].
All input matrices of genetic and geographic distance were constructed in GenAlEx v6.5 [50].

3.4 Evaluation of demographic history

To provide a temporal perspective on population demographics, extended Bayesian skyline (BS) and lineage through-
time (LTT-EBS) plots were constructed with Tracer v1.7.1 independently for the Pontocaspian and southern Iranian
*Theodoxus* groups. To eliminate potential bias of our interpretations, caused by individual phylogroups on the overall
trends, EBS plots were also constructed for individual phylogroups (Appendix A, figure S1). To generate the needed
input trees and log files, the BEAST v2.5.2 package and CIPRES Science Gateway were again used. The settings and
process followed similarly to the dated phylogeny described above (see 3.2 Phylogenetic analyses). However,
'Coalescent Extended Bayesian Skyline' was set at the tree prior, only in-group taxa of each group were used and
each EBS plot was generated through a single run of 800,000,000 MCMC generations (saving one tree and log in
every 20,000 generations and one EBSP log in every 5,000 generations). The best-fit models were again determined
by bModelTest differed (Pontocaspian group: COI = TN93, 16S = a variant of K81 with additional groups for the rate
r_{gt} (123323123323), ATP α = a variant of TN93TIM with additional groups for the rates r_{wg} and r_{gt} (121344123343);
southern Iranian group: COI = a variant of HKY with additional groups for the rate r_{gt} (121123), 16S and
ATP α = TN93). Dating followed a slightly different process: Crown nodes of each in-group were set according
to 95% confidence HPD intervals determined for the date of divergence of the most common recent ancestor
(MCRA), as established in the newly generated phylogeny below using a normally distributed prior (see figure 3).
Log files were assessed in Tracer v1.7.1 and only runs where all ESS values were > 200 were converted to EBS plots.
To construct the EBS plots, the output EBSP log files were imported into the BEAST package tool EBSPAnalyzer
v2.5.2 [37]. Linear reconstruction was selected and the burn-in removed was set as determined by parameter
convergence in Tracer v1.7.1.

4. Results

4.1 Phylogenetic structure

Based on the combined analyses, the monophyly of Pontocaspian and southern Iranian *Theodoxus* groups was well
supported (Posterior probability (PP) = 1.00; figure 3). Within the two groups, six phylogroups were identified (four
in southern Iran, termed I–IV herein, and two in the Caspian system, V–VI; figure 3). Three divergence events
occurred between southern Iranian phylogroups, while only a single divergence event occurred splitting the two
Caspian phylogroups (figure 3). The phylogeny suggests that the divergence of southern Iranian phylogroups from
one another begun earlier than the two Pontocaspian phylogroups (figure 3). Except for phylogroup III (with
specimens from Harat, Shahrabak, Qatruyeh), southern Iranian phylogroups are restricted to single localities.
However, there is both polyphyly and some parphyly between the morphospecies *T. pallidus* (Dunker, 1861) and *T.*
*doriae* Issel, 1865 (table 2; figure 3). In comparison, while there are haplotypes confined to specific locations among
the Pontocaspian *Theodoxus*, there are no location-specific phylogroups. Both supported phylogroups contain
individuals from various locations and similarly suggest parphyly and polyphyly among morphospecies commonly
identified as *T. major*, *T. schultzei*, *T. pallasi*, and *T. astrachanicus*, both within and between phylogroups (table 1;
figure 3).

4.2 Phylogeographic structure

TCS haplotype networks indicate less differentiation between haplotypes and more haplotype sharing among the Caspian localities when compared to those from southern Iran (figure 4). Haplotypic diversity (h) is high and similar for both groups, but on average it is marginally higher for the Caspian group (table 2). In comparison, nucleotide diversity is far lower in the Caspian group as compared to southern Iran, indicative of a younger flock (table 2). AMOVA analyses of each gene set indicate significant levels of regional population differentiation among some more distantly connected Pontocaspian localities (Appendix A, table S1). Similar AMOVA results were also found for southern Iran, although the results of the independent DNA datasets were more conserved (Appendix A, table S2). Significant IBD was found by the db-RDA within both groups, however in certain tests it was either not supported (i.e. the COI dataset) or was overall far weaker among Pontocaspian localities (table 2; Appendix A, figures S1 and S2B, supporting information).

4.3 Demographic history

The EBS plots suggest the effective population size (in respect to the generation time) in the Pontocaspian populations have remained stable until very recently (± 0.01 Ma) (c. 5 thousand years ago (ka)), while the southern Iranian group entered a steady demographic decline earlier, around 0.1 Ma (90 ka (figure 5a). The LTT plots demonstrate a relatively constant, nearly exponential pattern of lineage accumulation in the Pontocaspian group, compared to the more gradual and periodic, reverse sigmoidal, accumulation observed in the southern Iranian group. Individual EBS plots for the different phylogroups (I–VI; figure 3) of each of the two major groups did not all find suitable ESS values to warrant construction (potentially as a result of limited specimens in certain phylogroups) (figure S1). However, those that were supported (figure S1), show strong similarity to the combined EBS plots in figure 5. This similarity suggests individual phylogroups have had limited prejudice against the overall trends established for each group (figure 5b–c) and by combining data has only aided to increased support.

5. Discussion

5.1 Pontocaspian biogeography

We found some indication for an early catastrophic bottleneck reducing *Theodoxus* diversity to a single lineage in the Pontocaspian region during the Early Pleistocene. However, post this proposed bottleneck there is little intraspecific structure within the Pontocaspian *Theodoxus* group to warrant divergence events being specifically linked to environmental changes (figures 3, 4 and 5). In contrast to strong intraspecific phylogeographic structure and IBD observed in the southern Iranian *Theodoxus*, haplotypes belonging to the Pontocaspian *Theodoxus* group are shared among a variety of locations and IBD is either far weaker or not supported at all (table 2; figures 3 and 4; tables S1 and S2; figures S1 and S2). Therefore, our hypothesis that major low stands drove strong phylogeographic structure in the Pontocaspian *Theodoxus* group (as a result of populations that were confined to isolated refugia) is inconclusive prior to the Middle Pleistocene, but can be rejected from this point onward.

Supported intraspecific divergence events in Pontocaspian *Theodoxus* do not necessarily correspond with Caspian Sea low stands. Although the 95% confidence intervals on divergence events overlap with several low stands [8], they remain too broad to be attributed to a specific lake level or event (figure 3). Moreover, the BEES plot indicates a relatively stable population and LTT plot shows a nearly exponential accumulation of lineages post the mid-Bakunian until at least the Novocaspian stage (Holocene; figure 5a,b). If increased salinity and lake level changes 
[revised manuscript text omitted]

0.490 kMa may correlate with the onset of dryer climates and the first appearance of humans in the region [78–80].
While early humans probably had little environmental footprints as compared to today, an ever growing pressure for
water resources in conjunction with these climatic changes [80–82] may have posed increasing threats for *T. pallidus*.
As spring systems often contain unique and endemic haplotypes, dedicated conservation efforts of a variety of spring
systems are critical to protect this species (and potentially other aquatic species in southern Iran).

On the other hand, our analysis suggests that *T. major* has expanded over the last 105 kyr (figure 5a5). Considering
the entire range of the 95% confidence interval, however, both a population decline and increase is feasible. Already
for a number of Pontocaspian species population declines are evident [12]. Several species of Pontocaspian molluscs
are even thought to have gone extinct during the past century [12]. The causes for these declines are not well
established given the general lack of data for most species [12], but may be linked to anthropogenic activity [83].
However, the same causes could have been responsible for the apparent population increase, such as by reducing
competition. Consequently, species monitoring and distribution modelling are required to better assess its
conservation status.

6. Conclusion

Our analyses were able to detect strong phylogeographic structure in the southern Iranian group, compared to a
largely unstructured Pontocaspian *Theodoxus* group. This comparison indicates that the drivers of diversification in

[revised manuscript text omitted]

Data Accessibility

DNA sequences were deposited on Genbank, accessions numbers: still to add – will upload and amend internal tables with
accession numbers upon first acceptance of the research article, see temporary Appendix B (DNA sequence data) for the purposes
of the review process.

Voucher specimens of all processed material are stored in the Justus Liebig University Giessen, Systematics and Biodiversity
collection, Germany. DNA sequences were deposited on GenBank, accessions numbers: MN168547–MN168757, MN174926–
MN175136 and MN180417–MN180627. All nucleotide sequence alignment files used as input data for the various analyses,
output data used to construct the graphs of the EBS plot analyses and db-RDA input files with R script are available through Dryad
(<https://doi.org/10.5061/dryad.mn15f80>) [84].

Competing Interests

The authors declare no competing interests.

Authors' Contributions

All authors contributed to the conceptualisation of the research article. In particular A.F.S. conceptualised the study, A.F.S., S.N.,
375 M.F.H. and V.V.A. conducted the fieldwork and organised logistics, A.F.S. and S.N. performed the laboratory work, A.F.S. and
376 T.A.N. analysed the data and A.F.S., T.A.N., T.W. and C.A. leading the writing.

References

- 1. Bunje PME. 2007 Fluvial rang³⁸⁹ expansion, allopatry, and ³⁹⁰ parallel evolution in a Danubia³⁹¹ snail lineage (Neritidae): ³⁹² *Theodoxus*. *Biol. J. Linn. Soc* ³⁹³ 90, 603–617. ³⁹⁴ 2. (doi:10.1111/j.1095-8312.2007.00750.x) Bunje PME. 2005 Pan-

Commented [FS1]: Please note the temporary review link for review purposes:

<https://datadryad.org/review?doi=doi:10.5061/dryad.mn15f80>

European phylogeography of	480	5.035)	565	2008 Morphological and
the aquatic snail Theodoxus	481	Wesselingh FP et al. 2019	566	molecular genetic study of the
fluviatilis (Gastropoda:	482	Mollusc species from the	567	Persian sturgeon Acipenser
Neritidae). Mol. Ecol. 14 , 432	483	Pontocaspian region – an	568	persicus Borodin
4340. (doi:10.1111/j.1365-	484	expert opinion list. Zookeys	569	(Acipenseridae) taxonomic
294X.2005.02703.x)	485	827 , 31–124.	570	status. J. Ichthyol. 48 , 891–
3.	486	(doi:10.3897/zookeys.827.313	571	903.
Szabó K. 2009 An attempt to	487	5)	572	(doi:10.1134/S0032945208100
reveal the systematic	488	Audzijonyte A, Baltrūnaitė L,	573	068)
relationship between	489	Väinölä R, Arbačiauskas K.	574	22. Neubauer TA, Harzhauser M,
Theodoxus prevostianus (C.	490	2015 Migration and isolation	575	Kroh A, Georgopoulou E,
Pfeiffer, 1828) and Theodoxus	491	during the turbulent Ponto-	576	Mandic O. 2015 A gastropod-
danubialis (C. Pfeiffer, 1828)	492	Caspian Pleistocene create	577	based biogeographic scheme
(Mollusca, Gastropoda,	493	high diversity in the crustacea	578	for the European Neogene
Neritidae). Mollusca 27 , 95–	494	Paramysis lacustris . Mol. Ecol.	579	freshwater systems. Earth-
107.	495	24 , 4537–4555.	580	Science Rev. 143 , 98–116.
4.	496	(doi:10.1111/mec.13333)	581	(doi:10.1016/J.EARSCIREV.20
Sereda S V., Krizsik V. 2012	497	Audzijonyte A, Wittmann KJ,	582	15.01.010)
Extremely low genetic diversity	498	Ovcarenko I, Väinölä R. 2009	583	23. Neubauer TA, Harzhauser M,
in the endangered striped	499	Invasion phylogeography of the	584	Georgopoulou E, Kroh A,
nerite, Theodoxus transversalis	500	Ponto-Caspian crustacean	585	Mandic O. 2015 Tectonics,
(Mollusca, Gastropoda,	501	Limnomysis benedeni	586	climate, and the rise and
Neritidae) – a result of	502	dispersing across Europe.	587	demise of continental aquatic
ancestral or recent effects?	503	Divers. Distrib. 15 , 346–355.	588	species richness hotspots.
North. West. J. Zool. 8 , 300–	504	(doi:10.1111/j.1472-	589	Proc. Natl. Acad. Sci. U. S. A.
307.	505	4642.2008.00541.x)	590	112 , 11478–83.
5.	506	Audzijonyte A, Daneliya ME,	591	(doi:10.1073/pnas.1503992112
422	K. Schöll F, Thielsch A,	507	Väinölä R. 2006 Comparative	592)
Martens A. 2015 Theodoxus	508	phylogeography of Ponto-	593	24. Van Bocxlaer B. 2017
fluviatilis re-establishment in	509	Caspian mysid crustaceans:	594	Paleoecological insights from
the River Rhine: A native relic	510	isolation and exchange among	595	fossil freshwater mollusks of
or a cryptic invader? Conserv.	511	dynamic inland sea basins.	596	the Kanapoi Formation (Omo-
Genet. 16 , 247–251.	512	Mol. Ecol. 15 , 2969–2984.	597	Turkana Basin, Kenya). J.
(doi:10.1007/s10592-014-0665	513	(doi:10.1111/j.1365-	598	Hum. Evol.
7)	514	294X.2006.03018.x)	599	(doi:10.1016/J.JHEVOL.2017.0
6.	515	16. Brown JE, Stepien CA. 2008	600	5.008)
Stelbrink B, Neubauer TA,	516	Ancient divisions, recent	601	25. von Rintelen T, Stelbrink B,
Lazarev S, Wilke T, Albrecht	517	expansions: phylogeography	602	Marwoto RM, Glaubrecht M.
2019 Contributions of	518	and population genetics of the	603	2014 A snail perspective on the
biogeographical functions to	519	round goby Apollonia	604	biogeography of Sulawesi,
species accumulation may	520	melanostoma . Mol. Ecol. 17 ,	605	Indonesia: Origin and intra-
change over time in refugial	521	2598–2615.	606	island dispersal of the
regions. J. Biogeogr. 46 , 1274	522	(doi:10.1111/j.1365-	607	viviparous freshwater
1286.	523	294X.2008.03777.x)	608	gastropod Tylomelania . PLoS
7.	524	Cristescu MEA, Hebert PDN,	609	One 9 , e98917.
Brown JW, Keeble S, Hardwicke	525	Onciu TM. 2003	610	(doi:10.1371/journal.pone.0098
KM, Sullivan J, Harmon LJ.	526	Phylogeography of Ponto-	611	917)
2019 The choice of tree prior	527	Caspian crustaceans: a	612	26. Wilke T, Albrecht C,
and molecular clock does not	528	benthic-planktonic comparison	613	Anistratenko VV, Sahin SK,
substantially affect	529	Mol. Ecol. 12 , 985–996.	614	Yildirim MZ. 2007 Testing
phylogenetic inferences of	530	(doi:10.1046/j.1365-	615	biogeographical hypotheses in
diversification rates. PeerJ 7 ,	531	294X.2003.01801.x)	616	space and time: faunal
e6334.	532	18. Kotlík P, Marková S, Choleva	617	relationships of the putative
(doi:10.7717/peerj.6334)	533	L, Bogutskaya NG, Ekmekçi	618	ancient Lake Eğirdir in Asia
8.	534	FG, Ivanova PP. 2008	619	Minor. J. Biogeogr. 34 , 1807–
Quaternary time scales for the	535	Divergence with gene flow	620	1821. (doi:10.1111/j.1365-
Pontocaspian domain:	536	between Ponto-Caspian refugia	621	2699.2007.01727.x)
Interbasinal connectivity and	537	in an anadromous cyprinid	622	27. Bandel K. 2001 The history of
faunal evolution. Earth-Science	538	Rutilus frisii revealed by	623	Theodoxus and Neritina
Rev. 188 , 1–40.	539	multiple gene phylogeography	624	connected with description and
(doi:10.1016/J.EARSCIREV.2014	540	Mol. Ecol. 17 , 1076–1088.	625	systematic evaluation of related
18.10.013)	541	(doi:10.1111/j.1365-	626	Neritimorpha (Gastropoda).
9.	542	294X.2007.03638.x)	627	Mitteilungen aus dem Geol.
Chen JL, Pekker T, Wilson CR	543	19. Levin BA et al. 2017 Phylogen	628	Inst. der Univ. Hambg. 85 , 65–
Tapley BD, Kostianoy AG,	543	and phylogeography of the	629	164.
Cretau J-F, Safarov ES. 2015	544	roaches, genus Rutilus	630	28. Karpinsky MG. 2002 Ecology of
Long-term Caspian Sea level	545	(Cyprinidae), at the Eastern	631	the Benthos of the Middle and
change. Geophys. Res. Lett.	546	part of its range as inferred	632	Southern Caspian . Moscow:
44 , 6993–7001.	547	from mtDNA analysis.	633	VNIRO.
(doi:10.1002/2017GL073958)	548	Hydrobiologia 788 , 33–46.	634	29. Anistratenko VV, Zettler ML,
10. Kosarev AN. 2005 Physico-	549	(doi:10.1007/s10750-016-2984	635	Anistratenko OY. 2017 On the
geographical conditions of the	550	3)	636	taxonomic relationship between
Caspian Sea. In The Caspian	551	20. Parvizi E, Naderloo R,	637	Theodoxus pallasi and T.
Sea Environment (eds AG	552	Keikhosravi A, Solhjoui-Fard	638	astrachanicus (Gastropoda:
Kostianoy, AN Kosarev), pp.	553	S, Schubart CD. 2018 Multiple	639	Neritidae) from the Ponto-
31. Springer, Berlin,	554	Pleistocene refugia and	640	Caspian region. Arch. für
Heidelberg.	555	repeated phylogeographic	641	Molluskenkd. 146 , 213–226.
(doi:10.1007/978-3-642-002)	556	breaks in the southern Caspia	642	(doi:10.1127/arch.moll/146/213
11. Forte AM, Cowgill E. 2013 Late	557	Sea region: Insights from the	643	-226)
Cenozoic base-level variations	558	freshwater crab Potamon	644	30. Palumbi SR, Martin A, Romano
of the Caspian Sea: A review	559	ibericum . J. Biogeogr. 45 ,	645	S, McMillan WO, Stice L,
its history and proposed driving	560	1234–1245.	646	Grabowski G. 1991 The Simple
mechanisms. Palaeogeogr.	561	(doi:10.1111/jbi.13195)	647	Fool's Guide to PCR . Privately
Palaeoecol. 36 , 392–407.	563	21. Ruban GI, Kholodova M V.,	648	published document compiled
(doi:10.1016/J.PALAEO.2013.064	564	Kaimykov VA, Sorokin PA.	649	by S. Palumbi. Dept. Zoology,

[revised manuscript text omitted]

15X)	990	Expedition 1956): Beiträge zur
66. Zastrozhnov A et al. 2018	991	Kenntnis der Molluskenfauna
Biostratigraphical investigation	992	des Iran. Sitzungsberichte der
as a tool for	993	österreichischen Akad. der
palaeoenvironmental	994	Wissenschaften, Math. Klasse,
reconstruction of the	995	Abteilung I 166, 435–494.
Neopleistocene (Middle-Upper	996	76. Glöer P, Pešić V. 2012 The
Pleistocene) at Kosika, Lower	997	freshwater snails (Gastropoda)
Volga, Russia. Quat. Int.	998	of Iran, with descriptions of two
(doi:10.1016/J.QUAIN.2018.09	999	new genera and eight new
1.036)	1000	species. Zookeys 219, 11–61.
67. Abdullayev N. 2000 Seismic	1001	(doi:10.3897/zookeys.219.3406
stratigraphy of the Upper	1002)
Pliocene and Quaternary	1003	77. Dunker W. 1861 Beschreibung
deposits in the South Caspian	1004	neuer Mollusken. Malakozool.
Basin. J. Pet. Sci. Eng. 28, 1005	1005	Blätter 8, 35–45.
207–226. (doi:10.1016/S0920	1006	78. Sepehr M, Kharazian A, Nasab
4105(00)00079-6)	1007	HV, Jayez M, Hashemi S-M,
68. Eynoddin EH et al. 2017	1008	Nateghi A, Abdollahi A,
Assessment of relative active	1009	Kharazian MA, Berillon G. 2019
tectonics in the Bozghoush basin	1010	Anzo: The first evidence of
(SW of Caspian Sea). Open	1011	Paleolithic cave sites in the
Mar. Sci. 07, 211–237.	1012	northern margin of the Iranian
(doi:10.4236/ojms.2017.72016)	1013	Central Desert, Semnan, Iran.
69. Javadi Mosavi E, Arian M. 2011	1014	Archaeology 2019, 1–5.
Tectonic geomorphology of	1015	(doi:10.5923/j.archaeology.201
Atrak River, NE Iran. Open J	1016	90701.01)
Geol. 5, 106–114.	1017	79. Vahdati Nasab H, Hashemi M.
(doi:10.4236/ojg.2015.53010)	1018	2016 Playas and Middle
70. Sharkov E, Lebedev V,	1019	Paleolithic settlement of the
Chugaev A, Zabarinskaya L,	1020	Iranian Central Desert: The
Rodnikov A, Sergeeva N,	1021	discovery of the Chah-e Jam
Safonova I. 2015 The	1022	Middle Paleolithic site. Quat.
Caucasian-Arabian segment	1023	Int. 408, 140–152.
the Alpine-Himalayan	1024	(doi:10.1016/J.QUAIN.2015.1
collisional belt: Geology,	1025	1.117)
volcanism and neotectonics.	1026	80. Wang X, Wei H, Taheri M,
Geosci. Front. 6, 513–522.	1027	Khormali F, Danukalova G,
(doi:10.1016/J.GSF.2014.07.028	1028	Chen F. 2016 Early Pleistocene
1)	1029	climate in western arid central
71. Sokhadze G, Floyd M,	1030	Asia inferred from loess-
Godoladze T, King R, Cowgill	1031	palaeosol sequences. Sci. Rep.
ES, Javakhishvili Z, Hahubia	1032	6, 20560.
Reilinger R. 2018 Active	1033	(doi:10.1038/srep20560)
convergence between the	1034	81. Carolin S, Walker RT,
Lesser and Greater Caucasus	1035	Henderson GM, Maxfield L,
in Georgia: Constraints on the	1036	Ersek V, Sloan A, Talebian M,
tectonic evolution of the	1037	Fattahi M, Nezamdoust J. 2015
Lesser–Greater Caucasus	1038	Decadal-scale climate
continental collision. Earth	1039	variability on the Central Iranian
Planet. Sci. Lett. 481, 154–160	1040	Plateau spanning the so-called
(doi:10.1016/J.EPSL.2017.10.041	1041	4.2 ka BP drought event. Am.
07)	1042	Geophys. Union, Fall Meet.
72. Salzburger W, Van Bocxlaer	1043	2015, Abstr. id. PP21D-07
Cohen AS. 2014 Ecology and	1044	Zobeidi T, Yazdanpanah M,
evolution of the African Great	1045	Forouzani M, Khosravipour B.
Lakes and their faunas. Annu.	1046	2016 Climate change discourse
Rev. Ecol. Evol. Syst. 45, 419–447	1047	among Iranian farmers. Clim.
545. (doi:10.1146/annurev-	1048	Change 138, 521–535.
ecolsys-120213-091804)	1049	(doi:10.1007/s10584-016-1741-
73. Schultheiß R, Wilke T,	1050	y)
Jørgensen A, Albrecht C. 2011	1051	83. Lattuada M, Albrecht C, Wilke
The birth of an endemic	1052	T. 2019 Differential impact of
species flock: demographic	1053	anthropogenic pressures on
history of the Bellamyia group	1054	Caspian Sea ecoregions. Mar.
(Gastropoda, Viviparidae) in	1055	Pollut. Bull. 142, 274–281.
Lake Malawi. Biol. J. Linn. Soc.	1056	(doi:10.1016/J.MARPOLBUL.2
972	102, 130–143.	1057	019.03.046)
(doi:10.1111/j.1095-	1058	84. Sands AF, Neubauer TA,
8312.2010.01574.x)	1059	Nasibi S, Fasihi Harandi M,
74. Schultheiß R, Van Bocxlaer	1060	Anistratenko V V., Wilke T,
Wilke T, Albrecht C. 2009 Old	1061	Albrecht C. 2019 Data from:
fossils–young species:	1062	Old lake vs. young taxa: a
evolutionary history of an	1063	comparative phylogeographic
endemic gastropod	1064	perspective on the evolution of
assemblage in Lake Malawi.	1065	Caspian Sea gastropods
Proc. R. Soc. B Biol. Sci. 276,	1066	(Neritidae: Theodoxus). Dryad
2837–2846.	1067	Digital Repository.
(doi:10.1098/rspb.2009.0467)	1068	(doi:10.5061/dryad.mn15f80)
75. Starmühlner F, Edlauer A. 1969	1069
Ergebnisse der
Österreichischen Iran-
Expedition 1949/50 (Mit
Berücksichtigung der Ausbeute
der Österreichischen Iran-

Tables

Theodoxus species	Number of specimens	Locality	Country	GPS coordinates	GenBank accession numbers		
					COI	16S	ATPS ₄
Pontocaspian group							
T. schirazensis var. major	10	Yerevan	Armenia	40.16633° N, 44.48533° E	MN168547–MN168556	MN174926–MN174935	MN180417–MN180426
T. schirazensis var. major	5	Aknalich	Armenia	40.14288° N, 44.17117° E	MK754532–MK754534, MN168557–MN168558	MK754874–MK754876, MN174936–MN174937	MK755206–MK755208, MN180427–MN180428
T. pallasi	10	Blue Planet Beach	Azerbaijan	40.77513° N, 49.54489° E	MN168559–MN168568	MN174938–MN174947	MN180429–MN180438
T. pallasi	5	Pirallahli Island	Azerbaijan	40.489336° N, 50.330422° E	MK754691–MK754693, MN168569–MN168570	MK755030–MK755032, MN174948–MN174949	MK755347–MK755349, MN180439–MN180440
T. pallasi	10	Masalli	Azerbaijan	39.01879° N, 48.69972° E	MN168571–MN168580	MN174950–MN174959	MN180441–MN180450
T. cf. schirazensis var. major	10	Baba Aman Spring	Iran	37.48488° N, 57.43629° E	MN168581–MN168590	MN174960–MN174969	MN180451–MN180460
T. pallasi	5	Shahpol River	Iran	36.584867° N, 51.768145° E	MK754724–MK754725, MK754766	MK755063–MK755064, MK755105	MK755377–MK755378, MK755415
T. cf. schirazensis var. major	10	Zoeram Spring	Iran	37.31885° N, 57.73742° E	MN168591–MN168592, MN168593–MN168602	MN174970–MN174971, MN174972–MN174981	MN180461–MN180462, MN180463–MN180472
T. pallasi	10	Kuryk	Kazakhstan	43.183287° N, 51.652672° E	MN168638–MN168647	MN175017–MN175026	MN180508–MN180517
T. pallasi	10	Aktau	Kazakhstan	43.628058° N, 51.168252° E	MN168648–MN168657	MN175027–MN175036	MN180518–MN180527
T. pallasi	10	Saura Canyon	Kazakhstan	44.221987° N, 50.806791° E	MN168658–MN168667	MN175037–MN175046	MN180528–MN180537
T. schultzei	5	Caspian Sea	Kazakhstan	43.50589° N, 51.08473° E	MN168668–MN168672	MN175047–MN175051	MN180538–MN180542
T. astrachanicus	10	Danчих	Russia	46.34941° N, 48.01978° E	MN168673–MN168682	MN175052–MN175061	MN180543–MN180552
T. astrachanicus	10	Astrakhan	Russia	47.16708° N, 47.44868° E	MN168683–MN168692	MN175062–MN175071	MN180553–MN180562
T. astrachanicus	10	Selitrennoye	Russia	48.42905° N, 44.94628° E	MN168693–MN168702	MN175072–MN175081	MN180563–MN180572
T. astrachanicus	10	Volgograd	Russia	48.4541° N, 44.36181° E	MN168703–MN168712	MN175082–MN175091	MN180573–MN180582
T. astrachanicus	10	Volga–Don Canal	Russia	47.18499° N, 39.62985° E	MN168713–MN168722	MN175092–MN175101	MN180583–MN180592
T. astrachanicus	10	Rostov-on-Don	Russia	46.2857° N, 35.29080° E	MN168723–MN168732	MN175102–MN175111	MN180593–MN180602
T. astrachanicus	10	Utyukskij Liman A	Ukraine	46.14998° N, 35.04865° E	MN168733–MN168742	MN175112–MN175121	MN180603–MN180612
T. astrachanicus	5	Utyukskij Liman B	Ukraine	30.11875° N, 55.12171° E	MN168743–MN168747	MN175122–MN175126	MN180613–MN180617
Southern Iranian group							
T. doriae	10	Shahrababak	Iran	30.01509° N, 54.34030° E	MN168603–MN168612	MN174982–MN174991	MN180473–MN180482
T. doriae	10	Harat	Iran	29.169889° N, 54.684617° E	MN168613–MN168622	MN174992–MN175001	MN180483–MN180492
T. pallidus	10	Qatruyeh	Iran	27.82° N, 56.40° E	MN168623–MN168632	MN175002–MN175011	MN180493–MN180502
T. doriae	5	Poshtekeno Spring	Iran	30.66724° N, 52.29326° E	MN168633–MN168637	MN175012–MN175016	MN180503–MN180507
T. pallidus	10	Aspas	Iran	35.056477° N, 32.346027° E	MN168748–MN168757	MN175127–MN175136	MN180618–MN180627
Outgroups							
T. jordani	1	Bath of Aphrodite	Cyprus	41.1504° N, 22.52371° E	MK754676	MK755015	MK755332
T. transversalis	1	Vardar River	Macedonia		MK754769	MK755108	MK755416

Commented [F52]: Note the addition of GenBank numbers to the table

Population statistics		Pontocaspian group			Southern Iranian group		
		COI	16S	ATP α	COI	16S	ATP α
Number of specimens	n	175	175	175	45	45	45
Number of sequences	n	175	175	350	45	45	90
Haplotypic diversity	h	0.753	0.706	0.791	0.680	0.610	0.812
Nucleotide diversity	π	0.007	0.002	0.003	0.018	0.007	0.006
Isolation by distance (IBD)	R ²	0.035001	0.074008	0.024001	0.248180	0.441474	0.311058
	p value	0.019089	≤0.019000	≤0.019001	≤0.019001	≤0.019001	≤0.019001

Formatted: Line spacing: single

Figures

Figure and table captions

Table 1. Species details and GenBank accession numbers for Pontocaspian, southern Iranian and outgroup *Theodoxus* spp. Locality names correspond to those in figure 1.

Table 2. Comparative summary of population statistics for COI, 16S and ATP α datasets between Pontocaspian and southern Iranian *Theodoxus* groups. Note IBD was calculated using db-RDA (see 3.3 Phylogeographic structure).

Figure 1. Representative phenotypes of the Pontocaspian and southern Iranian *Theodoxus* species studied herein. Pontocaspian: (a, b) *T. pallasi* (UGSB 20712); (c, d) *T. astrachanicus* (UGSB 18130); (e, f) *T. pallasi* (23435); (g, h) *T. major* (24895); (i, j) *T. major* (24909); (k, l) *T. schultzi* (25116); (m, n). Southern Iranian: *T. doriae* (26038); (o, p) *T. pallidus* (26429). Scale bar = 1 mm.

Figure 2. Map depicting the locations of the sampling sites around the Pontocaspian system and southern Iran. Colours of dots correspond to the locations as indicated in the key. Dashed lines encircle (I) the Pontocaspian and (II) the southern Iranian *Theodoxus* sampling localities. The size of the dots represent the sample size at each location (larger = 10; smaller = 5 specimens).

Figure 3. Dated phylogeny of Pontocaspian and southern Iranian *Theodoxus* spp. constructed in BEAST based on COI, 16S and ATP α sequence data. Supported phylogroups of Pontocaspian and southern Iranian *Theodoxus* are labelled I to VI. Node labels among these phylogroups and outgroup species denote divergence time in millions of years ago (Ma), with the 95% credibility interval given in parentheses and as grey bars for in-group taxa. Small red squares at nodes (with darkened node bars and, in some instances, dates) indicate significant posterior probabilities of divergence events. Parallel to each supported phylogroup, coloured bars indicate the localities and respective morphospecies of the included specimens as defined in the key on the left (also see figure 2). Caspian Sea lake level variations over the last 1.5 Ma (relative to absolute sea level) and regional stratigraphy (following the “short-Akchagylian” option) are adapted from Krijgsman et al. [8] (Khv. = Khvalynian).

Figure 4. Statistical haplotype networks for COI, 16S and ATP α sequence data for Pontocaspian and southern Iranian *Theodoxus* groups. The total number of sequences in each network are demarcated by ‘n’. The circle sizes represent relative frequency of sequences per haplotype. The number of site changes separating haplotypes are indicated by blank dots. Colours correspond to the sampling locations as indicated in the key and in figure 2. Haplotype groupings are boxed and labelled according to phylogroups determined through the dated phylogeny (I–VI; figure 3).

Figure 5. a) Extended Bayesian skyline (BSEBS) plots indicating population trends for the Pontocaspian (blue) and southern Iranian (red) *Theodoxus* groups. b) Lineage through time (LTT) plots indicating the accumulation of lineages through time for Pontocaspian and southern Iranian *Theodoxus*. In the BS plots, the central line of each plot represents the median value, while in the LTT plots the line rather constitutes the mean. In both plots and the shaded area indicates the 95% confidence interval and floating grey bars indicate the 95% confidence interval associated with the onset of intraspecific diversification. The LTT and BS plots indicate respectively. Note the EBS plots depict marginally younger intraspecific diversity different starting dates for each group when compared with the phylogeny. Importantly however, there is strong overlap in of EBS plot starting dates with the 95% confidence intervals HPDs established among the analyses for the onset of intraspecific diversification in each group as shown in the phylogeny (figure 3).

Appendix B

To the Chief Editor, Royal Society Open Science

Dear Prof. Jeremy Sanders CBE FRS

Cc: Prof. Kristina Sefc (Associate Editor), Prof. Kevin Padian (Subject Editor)

Reference Number: RSOS-190965

We very much appreciate the time and effort of the reviewer and the editors in assessing our revised paper and we are grateful for their additional comments. Below we list all critical comments and suggestions of the associate editor and reviewer (*in italics*), together with our point-by-point replies (*in blue*).

ASSOCIATE EDITOR (Dr Kristina Sefc):

I'd like to thank you for addressing most of the concerns raised in the first round of review. The issue of pooling divergent lineages for the EBSPs still remains and actually represents a rather serious one since an important conclusion - regarding the decline in population size - may be based on an artefact in the analysis. Please see the reviewer's comments for details. If the sampling / the data don't allow to analyse the demographic history of the Iranian populations, then you'll have to consider dropping this part from the manuscript (or interpret the network structures of the mtDNA and ncDNA sequences verbally - less diversity in the mt than nc genomes within each phylogroup may indeed point to a recent bottleneck); or at least discuss the problem associated with pooling the divergent lineages and make clear that there's a risk of a spurious result.

We note the feedback of the reviewer and the associate editor regarding the EBSPs. We understand there may still be a potential bias from pooling the Iranian data without being able to assess the EBSP trends in individual Iranian phylogroups due to the ESS values lacking support (potentially as a result of small sample sizes of the phylogroups). We have thus followed the associate editor's suggestions:

1) We have added further interpretation of our networks and haplotypic diversity to add support for a bottleneck within the Iranian species as observed in the EBSP (see L171-174, L190-191 & L284-286).

2) We now note that "caution" should be taken with the Iranian group's EBSP when interpreting the trend of the pooled data (figure 5) given the bias that may be caused (L190-192, L282-284 & L289-290). We reference the papers noted by the **reviewer** in this regard (L135 & L283-284). However, we also note that Heller et al. (2013) indicates that at least in part some of these concerns may be mitigated by a balanced sampling strategy such as our own (L284). Moreover we note that even neglecting the EBSP (from a conservation perspective), given current climate shifts and anthropogenic threats to spring habitats in the region, conservation efforts may still be needed to protect this species (L290-293).

REVIEWER: 1

MAJOR COMMENTS

1. Lines 175-177 (and comment 10 of the original review): Note that each node in the tree represents a divergence event. In fact, you can delete this entire sentence as it essentially conveys the same information as the previous sentence (lines 174-175).

We have removed this sentence as suggested.

2. EBSPs: Yes, doing separate analyses for the two Pontocaspian phylogroups (now presented as Supplementary info) is the right way to deal with these data. And yes, for most of the Iranian phylogroups sample size/diversity is too small to allow for doing EBSP analyses. This, however, is no justification for pooling all these four pretty divergent phylogroups (cryptic species?) for a single EBSP analysis. You have virtually no intraphylogroup diversity in the Iranian samples and with the large interphylogroup divergence you inevitably get a signature of a drastic recent population decline in your EBSP. It's simply not possible to do reliable EBSP analyses with these Iranian samples. I suggest you have a look at previous studies that evaluated the performance of (E)BSPs in the presence of population structure. Specifically, in highly structured populations, drastic recent population size declines were observed (similar to what you found with your data) even though the sequences were simulated under a constant population size scenario. Two relevant studies that come to my mind are Heller et al. 2013 PLoS One & Grant 2015 JHered. I think you'll have to have a more critical look at your EBSPs, especially since these data are essential for your discussion/conclusions on conservation aspects in your gastropods.

Please see response to the associate editor's comment.